# Ghost-arc geochemical anomaly at a spreading ridge caused by supersized flat subduction

Guido M. Gianni [1,2], Jeremías Likerman [2,3], César R. Navarrete [2,4,7], Conrado R. Gianni[1,7] & Sergio Zlotnik [5,6,7] ✉

The Southern Atlantic-Southwest Indian ridges (SASWIR) host mid-ocean ridge basalts with a residual subduction-related geochemical fingerprint (i.e., a ghost-arc signature) of unclear origin. Here, we show through an analysis of plate kinematic reconstructions and seismic tomography models that the SASWIR subduction-modified mantle source formed in the Jurassic close to the Georgia Islands slab (GI) and remained near-stationary in the mantle reference frame. In this analysis, the GI lies far inboard the Jurassic Patagonian-Antarctic Peninsula active margin. This was formerly attributed to a large-scale flat subduction event in the Late Triassic-Early Jurassic. We propose that during this flat slab stage, the subduction-modified mantle areas beneath the Meso-zoic active margin and surrounding sutures zones may have been bulldozed inland by >2280 km. After the demise of the flat slab, this mantle anomaly remained near-stationary and was sampled by the Karoo mantle plume 183 Million years (Myr) ago and again since 55 Myr ago by the SASWIR. We refer to this process as asthenospheric anomaly telescoping. This study provides a hitherto unrecognized geodynamic effect of flat subduction, the viability of which we support through numerical modeling.

Mid-ocean ridges (MOR) form the most extensive plate boundary network concentrating seismicity and massive magmatism associated with the extrusion of mid-ocean ridge basalts (MORB)[1,2]. During their lifetime, MORs remain stationary or migrate relative to the mantle, building oceanic crust through the cooling of melts derived from variably depleted and less frequently enriched upper mantle sources[2–7]. In the last decades, a growing database of surprising geo-chemical signatures has suggested that the mantle below some mod-ern MORs indicates ongoing or former interaction with mantle plumes or delaminated ancient continental lithosphere[3,8–10]. More rarely, pre-vious studies have documented radiogenic isotopic abundances and

trace elements ratios indicative of a residual geochemical subduction-related signature (e.g., low Ce/Pb, low Nb/U, high Nb/Zr, high Ba/La, high $H_2O$/Ce), thought to result from the introduction into the con-vecting mantle of continental or pelagic sediments, altered oceanic crust, and/or metasomatic slab-derived fluids linked to former sub-duction zones (e.g.,[7,11–20]). This geochemical imprint is more subtle than those found in melts derived from a backarc basin mantle and is referred to as a "ghost-arc"[16] or "backarc basin basalt-like"[19] geo-chemical signature. While mantle plume-MOR interactions are rela-tively well-understood, from a numerical and plate kinematic perspective[10,21,22], how parcels of residual subduction-modified mantle

[1]Instituto Geofísico Sismológico Ing. Fernando Volponi (IGSV), Universidad Nacional de San Juan, San Juan, Argentina. [2]National Scientific and Technical Research Council (CONICET), Capital Federal, Argentina. [3]Instituto de Estudios Andinos Don Pablo Groeber, Universidad de Buenos Aires, Capital Federal, Argentina. [4]Laboratorio Patagónico de Petro-Tectónica, Universidad Nacional de la Patagonia "San Juan Bosco", Comodoro Rivadavia, Chubut, Argentina. [5]Laboratori de Càlcul Numéric, Escola Técnica Superior d'Enginyers de Camins, Canals i Ports, Universitat Politécnica de Catalunya, Barcelona, Spain. [6]Centre Internacional de Métodes Numérics a l'Enginyeria (CIMNE), Barcelona, Spain. [7]These authors contributed equally: César R. Navarrete, Conrado R. Gianni, Sergio Zlotnik. ✉e-mail: sergio.zlotnik@upc.edu

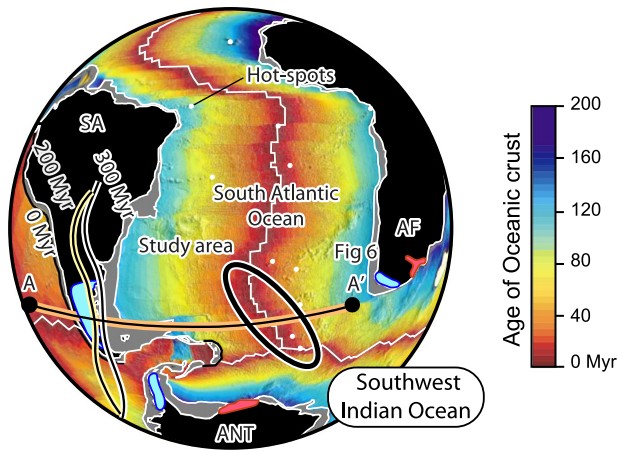

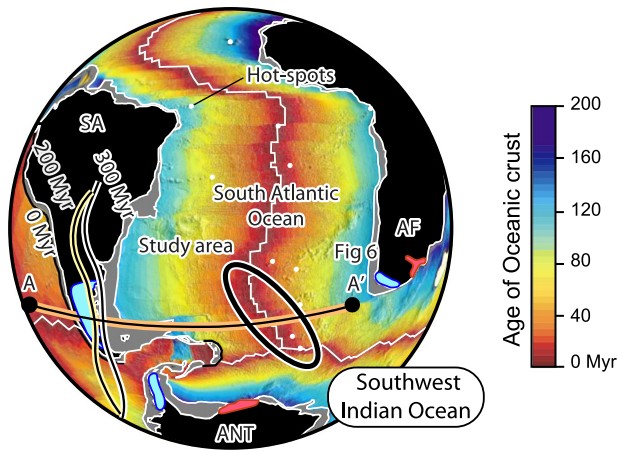

🔴 Subduction-related domain of the Karoo magmatic province [25]

🔵 Tectonomagmatic record of the South Gondwana flat-slab [42]

**Fig. 1 | Tectonic setting of the Southern Atlantic and Southwestern Indian oceans.** Map indicating the location of the Southern Atlantic-Southwest Indian ridges subduction-modified mantle[12,19], paleo-trench locations at 200 and 300 Myr ago[34], the tectonomagmatic record of the South Gondwana flat slab[42], and the subduction-related domain of the Southern Karoo magmatic province[25]. Age grid is taken from the GPlates Portal (https://www.portal.gplates.org/). *SA* South American plate, *AF* African plate, and *ANT* Antarctic Plate.

occur underneath MORs remains unclear. Early attempts to explain such occurrences consider that melting of a mantle source, initially entrained with slab fluids and sediments in the mantle wedge, forms refractory arc mantle relics that are dragged away from subduction zones by upper mantle convection to be later emplaced beneath MORs[7,11,12,19]. Another prominent hypothesis suggests that the upper mantle preserves geochemical memory of past convergent processes, where wide zones of subduction-modified asthenosphere remain near-stationary in the mantle reference frame through time [>100 million years (Myr) ago][13,16,17]. The preservation potential of these mantle source anomalies has been recently explained by the presence of relict mélange diapirs[16] consisting of hydrous subducted sediments and depleted mantle peridotites ascending into the mantle above the subducting slab by positive buoyancy in the arc and back-arc regions[23]. According to Richter et al.[16], after subduction and mantle wedge melting ceased, the mélange remnants would remain neutrally buoyant in the area due to compositional and thermal re-equilibration of density differences with the ambient mantle. These stagnant mantle anomalies are then sampled during MOR overriding above these areas[13,14,16] or by other processes, such as plowing by continental edges[17], or even during mantle plume impingement[24].

In this study, we address the origin of a localized MOR area in the Southern Atlantic-Southwest Indian ridges (SASWIR) with a conspicuous ghost-arc geochemical signature[12,19] that we refer to as the SASWIR subduction-modified mantle anomaly (Fig. 1; Supplementary Fig. 1; Supplementary Data file 1). The origin of the SASWIR subduction-modified mantle has been previously linked to Mesozoic convergence in the active margin of Southwestern Gondwana[12,19]. In that hypothesis, an asthenospheric flow would have transported ambient subduction-modified mantle from the Mesozoic active margin of southwest Gondwana towards the MOR that also interacted with ascending mantle plumes and delaminated mantle fragments beneath this region[12]. According to Le Roux et al.[12], onshore intraplate magmatism with subduction-related signatures documented in the Southern domain of the Karoo large igneous province (ca. 186-180 Myr ago) in South Africa and Northern Antarctica (e.g.,[25]) provides evidence for the

presence of a subduction-modified mantle beneath the study area at least since the Early Jurassic[12]. To date, exactly how upper mantle convection transports sources with ghost-arc geochemical signatures to MOR areas is still unclear, particularly in light of global seismic tomography observations discarding layered mantle convection (e.g.,[26]). Also, a subduction-modified mantle beneath the study area since at least the Early Jurassic[12] seems more compatible with a model involving a long-lived near-stationary geochemical anomaly[13,14,16,17]. More importantly, upper mantle convection carrying residual subduction-modified asthenosphere from the South American and Antarctic Peninsula active margins to MOR regions struggles to explain the restricted character of the SASWIR subduction-modified mantle source (Supplementary Fig. 1). To assess the origin of the SASWIR subduction-modified mantle anomaly, we first reconstruct the position of the subduction-related Southern Karoo magmatic province in the mantle reference frame. With this analysis, we evaluate the proposed link with the SASWIR subduction-modified mantle anomaly[12] and infer how long the anomaly could have resided beneath the study area. Then, we examine the mantle through global seismic tomography models to search for a potential fossil slab that could explain the existence of the SASWIR subduction-modified mantle anomaly (e.g.,[16,17]). Also, we couple seismic tomography sections and plate kinematic reconstructions assuming average slab sinking rates to link the mantle slab record to the reconstructed active margin in different mantle reference frames. Based on these reconstructions and a new 2-D thermomechanical modeling, we argue that an anomalously large flat subduction event forced an inland transport of previously metasomatized asthenosphere, placing subduction-modified mantle in an intraplate area later sampled by the SASWIR.

## Results

### Linking the SASWIR subduction-modified mantle to the Mesozoic South Gondwana flat slab

A subduction influence in MORBs from the SASWIR is evidenced when compared with samples from the Pacific Ocean ridge, holding a mantle source almost isolated from the effects of neighboring subduction zones[19] (Fig. 2; Supplementary Data file 2). In terms of composition, the SASWIR is characterized by sub-alkaline basalts with minor basaltic andesites and alkaline trachy-basalts, while the Pacific Ocean ridge shows a wider spectrum towards more evolved compositions (basaltic-andesites and andesites) (Fig. 2a). The ghost-arc geochemical signature in the SASWIR manifests when analyzing trace elements ratios sensitive to the influence of subduction-derived components (e.g.,[27,28] and references therein). The Ce/Pb ratio is lower in the SASWIR than in the Pacific Ocean ridge (SASWIR averages: 21.99 and 22.768-Pacific Ocean ridge average: 26), while the Ba/La (SASWIR averages: 7.86 and 7.27-Pacific Ocean ridge average: 3.57), Th/La (SASWIR averages: 0.099 and 0.074-Pacific Ocean ridge average: 0.052), and Ba/Nb ratios (SASWIR averages: 6.09 and 8.24-Pacific Ocean ridge average: 4.05) are higher in the former (Fig. 2b–f). These values support the influence of subduction-derived components (fluids ± sediments) previously suggested in the SASWIR[12,19]. Entrainment of sediments in the SASWIR mantle source is also evidenced by the recent finding of 2.8 Ga zircon xenocrysts of continental origin hosted in 53-35 Myr MORBs from the Shaka fracture zone of the Southwest Indian ridge[29].

For our analysis, we mapped the low-velocity perturbations of dv/v −1 and −2% at a depth of 100 km from the regional SA2019S high-resolution S-wave seismic tomography model of Celli et al.[30]. We use these low-velocity anomalies as a reference for the active mantle source region beneath the SASWIR (Fig. 3a; Supplementary Fig. 2). Also, we carried out a reconstruction at 183 Myr ago of the subduction-related domain of the Southern Karoo intraplate magmatic province (186-176 Myr ago)[25,31] considering plate kinematic models that apply different mantle reference frames[32–35] (Methods). We observe that the

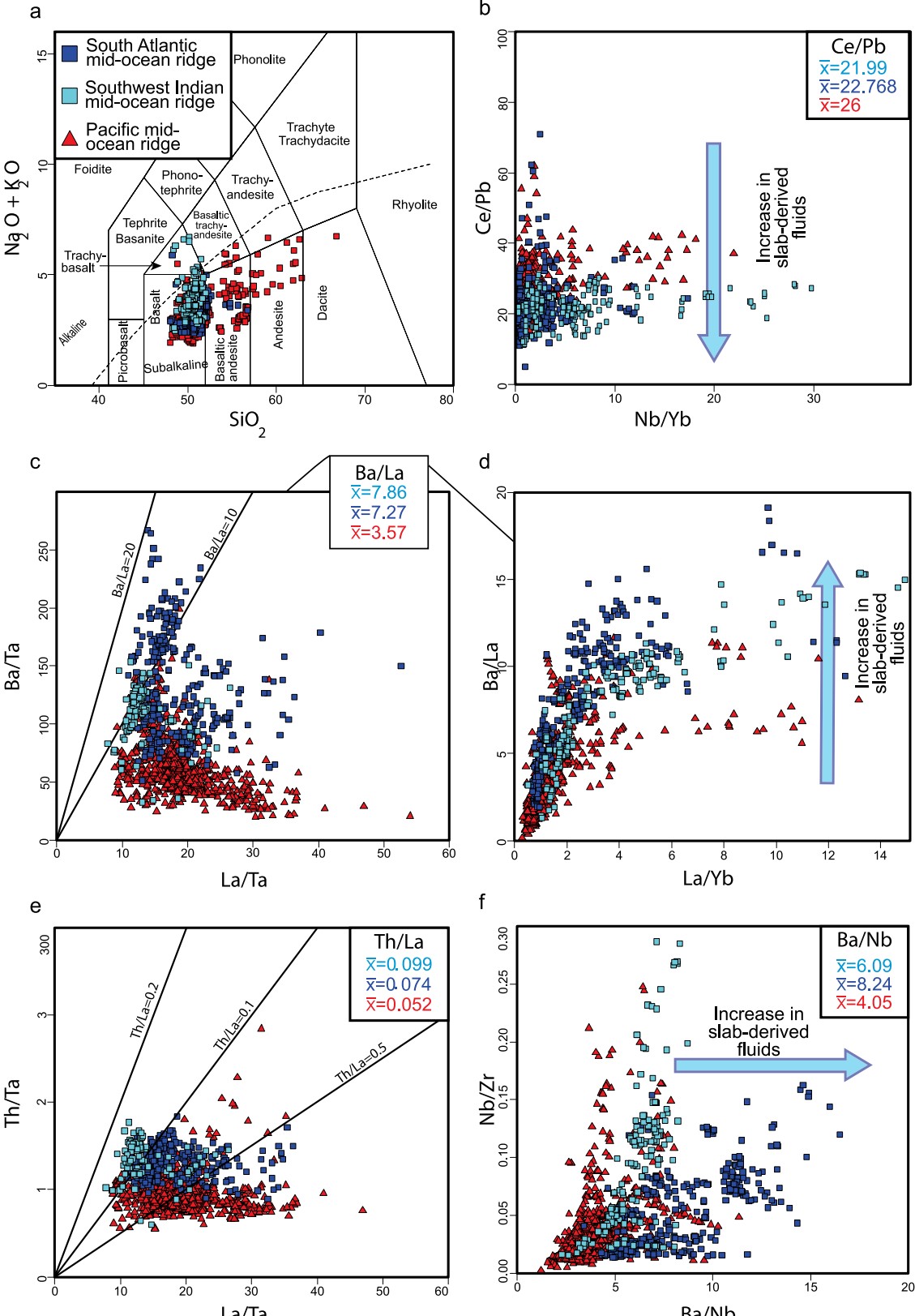

**Fig. 2 | Total-alkali-silica diagram and trace-element systematics of mid-ocean ridge (MOR) basalts from the Pacific, Southern South Atlantic, and Southwest Indian MORs.** Plots of **a** Na₂0+K₂0 vs. Si₂0, **b** Ce/Pb vs. Nb/Yb, **c** Ba/Ta vs. La/Ta, **d** Ba/La vs. La/Yb, **e** Th/Ta vs. La/Ta, **f** and Nb/Zr vs. Ba/Nb for the Pacific, South Atlantic (33°-55°S), and Southwest Indian (54° 1' 48"S, 3° 31' 48"E; 52° 51' 36"S, 19° 54'

36"E) MOR basalts illustrating the slab-derived influence of the Southern Atlantic-Southwest Indian ridges documented by Yang et al.[19] when compared to samples from the Pacific Ocean MOR. Geochemical data is available in Supplementary Data file 2.

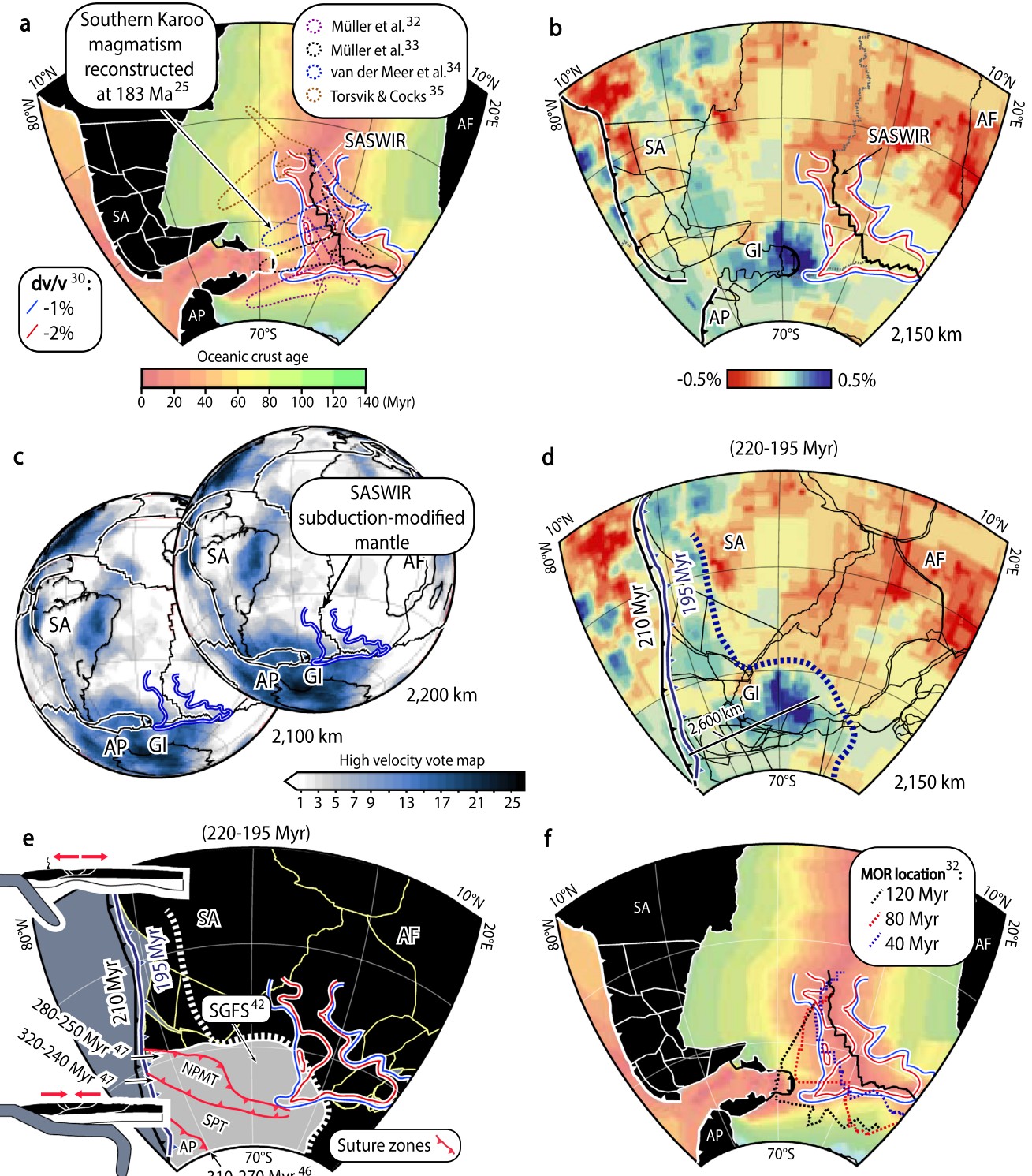

**Fig. 3 | Reconstructions linking the Southern Atlantic-Southwest Indian ridges (SASWIR) subduction-modified mantle anomaly, the Southern Karoo magmatic province, and the mantle structure beneath the South Gondwana margin. a** Reconstructions of the subduction-influenced Southern Karoo intraplate magmatic province[25] at 183 Myr ago showing a close spatial relationship to the SASWIR subduction-modified mantle zone indicated by low-velocity anomalies from the SA2019S S-wave seismic tomography model[30]. **b** Mantle structure at 2150 km[36] beneath the study area showing the SASWIR subduction-modified mantle zone positioned to the east of the Georgia Islands slab. **c** High-velocity vote maps confirming the spatial relationship between the Georgia Islands slab and the SAS-WIR subduction-modified mantle. **d** Reconstruction considering a whole-mantle

slab sinking rate of 1.1 cm/yr[37] linking the mantle structure at a depth of 2150 km and a plate kinematic reconstructions at 195 Myr ago[32]. Trench at 210 Myr ago is also shown in black. This reconstruction shows the SASWIR subduction-modified mantle positioned in front of a landward offset of high-velocity anomalies associated with the presence of the Georgia Island slab[38], previously interpreted as the mantle record of the South Gondwana flat slab[40,42]. **e** Interpretation of **d** including late Paleozoic suture zones in Patagonia[46,47] that preceded the flat slab. **f** mid-ocean ridge migration since 120 Myr ago[32] over the SASWIR subduction-modified mantle zone. SA South American plate, AF African plate; AP Antarctic Peninsula, SPT South Patagonian terrane, NPMT North Patagonian Massif terrane, SGFS South Gondwana flat slab, and GI Georgia Islands slab.

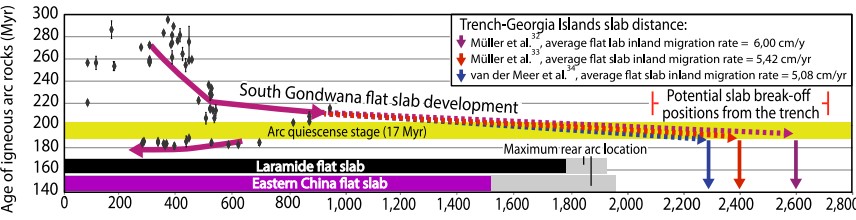

**Fig. 4 | Spatio-temporal diagram of the Mesozoic South Gondwana magmatism indicating possible average flat slab tip migration rates.** Arc rock age distance to paleo-trench vs. time[40] used to infer the flat slab tip kinematics before slab break-off at ca. 185 Myr ago[38,42] applying plate kinematic reconstructions with different mantle reference frames[32–34]. Inferred flat slab extent from the Mesozoic Laramide[50] and Eastern China[49] large-scale flat slab events are shown for comparison with potential sizes of the South Gondwana flat slab. Reconstructions considering the mantle reference frame of Torsvik and Cocks[35] were excluded from this analysis because yield results not compatible with the geological records of the western margin of Gondwana (see Results section). Geochronological data is available in Supplementary Data file 3.

Southern Karoo magmatic province[25] was emplaced in an area overlapping the current location of the SASWIR subduction-modified mantle zone (Fig. 3a). This observation lends support to the proposal of a long-lived nature of the subduction-modified mantle beneath the SASWIR[12].

To evaluate a potential link between the SASWIR subduction-modified mantle anomaly and ancient slab records[16,17], we inspected the mantle structure beneath the study area utilizing the UU-P07 P-wave global seismic tomography model[36] (Fig. 3b). To identify a consistent lower mantle structure across several global seismic tomography models, we built high-velocity vote maps considering 26 different P- and S-wave global seismic tomography models (Methods; Supplementary Table 1). These maps are generated by stacking a series of seismic tomography models at a specified mantle depth and detecting where the models agree based on an increasing vote count[37]. We note that the SASWIR subduction-modified mantle zone is positioned immediately to the east of a high-velocity perturbation in the lower mantle referred to as the Georgia Islands anomaly[38], which is shown at 2150 km and deeper (Fig. 3b, c). This mantle anomaly has a wall-like geometry extending at depths from about 1700 km to 2800 km and is interpreted as a fossil slab recording Early Jurassic subduction at depths analyzed in this study[39,40] and Permian subduction at the base[38,41]. An even closer spatial relationship between the Georgia Islands slab and the SASWIR subduction-modified anomaly is observed in the vote maps comparing multiple P- and S-wave seismic tomography models (Fig. 3c).

An analysis coupling the seismic tomographic slice at 2150 km with a recent Mesozoic plate kinematic reconstruction[32] depicts the trench position and the subduction mantle record at ca. 195 Myr ago assuming an average slab sinking rate of 1.1 cm/yr[37,38] (Fig. 3d) (Methods). This reconstruction indicates that high-velocity anomalies form a fringe that roughly aligns with the active margin of Southwest Gondwana, supporting the interpretation of a Jurassic subduction record[38] (Fig. 3d). Nevertheless, the Georgia Islands slab lies far inboard of the reconstructed active margin compared to high-velocity anomalies to the north (Fig. 3d). This observation, along with the geological record of inboard migration of the magmatic arc, subsequent arc shut-off, and compression between ca. 220 and 185 Myr ago in Patagonia, Antarctic Peninsula, and South Africa, led to the proposal of a flat subduction event referred to as the South Gondwana flat slab[40,42–44] (Figs. 1 and 3e). The demise of this subduction configuration took place at ca. 185-180 Myr ago during a slab break-off event that detached the Georgia Islands slab at the leading edge of the flat slab[42].

This spatio-temporal relationship between the mantle and surface geological records is also observed in plate kinematic reconstructions applying alternative mantle reference frames[32–34] (Supplementary Fig. 3a, b). The only exception is the reconstruction including the mantle reference frame of Torsvik and Cocks[35]. This analysis presents similar results to the rest of the reconstructions at 220-210 Myr ago but departs at 195 Myr ago, depicting major offsets between the mantle slabs and the reconstructed trench (Supplementary Fig. 3c). This observation could be interpreted as indicating Early Jurassic intra-oceanic subduction to the west of Southwestern Gondwana. Nevertheless, this scenario is not compatible with the geological records of an Early Jurassic continental arc-back arc system along Western South America[40,42,43]. Thus, our reconstructions show that the SASWIR subduction-modified mantle zone is likely an ancient feature formed in close spatial relationship to the Early Jurassic subduction of the Georgia Islands slab (Fig. 3a–e; Supplementary Fig. 3).

As the Georgia Islands slab indicates the maximum inland extent of the South Gondwana flat-slab before its demise[42], this paleogeographic coincidence possibly implies a genetic connection between the location of the SASWIR subduction-modified mantle zone and the Mesozoic flat slab process. Due to the relatively stationary character of Pangea at this time[32–34] (Figs. 1 and 3d, e; Supplementary Fig. 3a, b), the extreme inland growth of the South Gondwana flat slab would not have been caused by forced trench retreat as documented in active flat subduction in the Andes[45]. Thus, this process most likely resulted from large-scale inland migration of the flat slab tip at different possible rates that, depending on the mantle reference frame used, are 6 cm/yr[32], 5.42 cm/yr[33], and 5.08 cm/yr[34] (Fig. 4, Methods). We suggest that the development of the South Gondwana flat slab must have shifted frontally subduction-modified asthenosphere beyond the Georgia Islands slab beneath the Gondwana interior (Fig. 3e). As the flat slab followed the final assembly of Patagonia by 60-20 Myr[46,47], this process probably collected both the subduction-modified mantle beneath the arc and backarc regions as well as that present in the vicinity of suture zones beneath intraplate areas in Patagonia and South Africa (Fig. 3e). Thus, flat subduction would have led to the accumulation of an upper mantle with slab-derived components far inboard the Jurassic margin giving place to the SASWIR subduction-modified mantle anomaly beneath the study area. The SASWIR likely started to sample this long-lived near-stationary subduction-modified mantle in the last 50-35 Myr as it migrated progressively to the east during the Cretaceous-Cenozoic[32,48] (Fig. 3f).

## Numerical modeling of large-scale flat subduction and related mantle flow

The inland growth of the South Gondwana flat slab would have taken place with different possible magnitudes depending on the mantle reference frame employed. The reconstructions yield variable but large paleo-trench-slab distances (2600 km[32], 2400 km[33], and 2280 km[34]) (Figs. 3d, 4, Supplementary Fig. 3a, b). We note that these potential flat slab sizes exceed those associated with large-scale cases

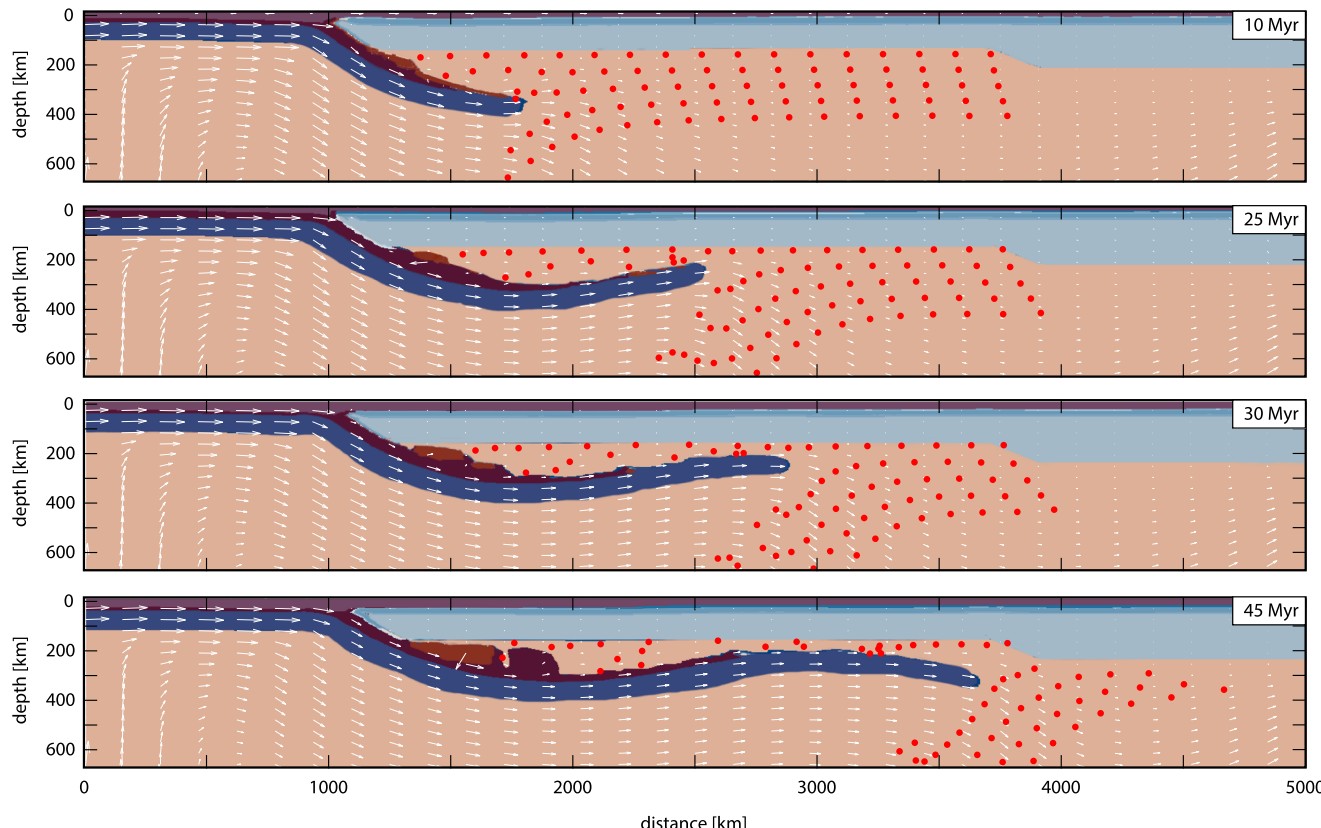

**Fig. 5 | 2-D thermomechanical numerical modeling of large-scale flat subduction and related upper mantle flow.** Model evolution showing the development of a large-scale flat-slab of about 2700 km in 35–40 Myr. The progressive deformation of the Lagrangian grid (red markers) illustrates the lateral shifting of the upper mantle during the inland growth of the large-scale flat subduction. Material colors as in Supplementary Data Fig. 4; white lines are velocity vectors. The maximum velocity in all panels is 8.4 cm/yr.

documented in Eastern China[49] and Western North America[50,51] (Fig. 4). The inland flat slab extents associated with these events would not have surpassed ~1900–1500 km considering average arc-trench distances (~287 km, https://www.earthbyte.org/calculating-arc-trench-distances-using-the-smithsonian-global-volcanism-project-database/). Thus, the South Gondwana flat slab, with a potential inland extent between ~2280 and 2600 km, constitutes the largest flat subduction episode documented so far, an aspect largely overlooked in previous studies[40,42].

We use these correlations in a 2-D numerical geodynamic modeling to infer the conditions under which the flat subduction and the forced mantle flow processes occurred (Methods, Supplementary Fig. 4). Note that the numerical models do not intend to reproduce complex physical (e.g., buoyant melange diapirs) nor geochemical processes involved in the mantle wedge but to inform about the mantle dynamics during large-scale flat subduction (see Methods). For simplicity in our model, we assume no trench-parallel mantle flow. Nevertheless, some mantle flow of this kind during the flat slab process must have taken place and could be the cause of the SASWIR subduction-modified mantle anomaly exceeding the lateral edges of the Georgia Island slab (Fig. 3d). In this model, a large-scale flat subduction geometry is developed for 2600–2700 km in ca. 35-40 Myr, which is about the life-span of the South Gondwana flat slab (Figs. 4 and 5). Also, this model effectively reproduces a large-scale frontal translation of the upper mantle during inland growth of the flat slab (Fig. 5).

## Discussion

The restricted and long-lived character of the SASWIR subduction-modified mantle is challenging to explain from a geodynamic point of view. As Mesozoic-Cenozoic subduction took place along the entire Western South American and Antarctic Peninsula margin[32,34], a relatively homogeneous occurrence of ghost-arc geochemical signatures in the South Atlantic MOR would be predicted if caused by upper mantle convection[7,19]. Instead, MORB samples with ghost-arc signatures are scattered along the Atlantic MOR compared to the strong clustering within our study area (Supplementary Fig. 1). Also, this model implies an eastward upper mantle flow of subduction-influenced asthenosphere that opposes continental plate motion in the study area, which is challenging to explain from a geodynamic point of view. An alternative explanation could be the melting of delaminated mantle lithosphere metasomatized by slab fluids present in the MOR source[3,9,52]. However, no large-scale high-velocity anomalies have been detected beneath the study area, precluding the presence of large blocks of metasomatized delaminated lithosphere as the main cause of ghost-arc geochemical signatures in the SASWIR (Supplementary Fig. 5). Another possibility could be a westward mantle plume flow towards the MOR[8] emplacing metasomatized cratonic lithosphere fragments from the African plate, a process which would be difficult to resolve in seismic tomography models. This mechanism was recently invoked to explain extremely low $^{187}Os/^{188}Os$ ratios in localized samples in the southernmost area of the Southwestern Indian MOR[10]. However, this process is difficult to generalize to the whole SASWIR subduction-modified mantle source, which extends for about 4400 km (Supplementary Fig. 1). An alternative explanation could be the development of a long-lived stationary geochemical anomaly as recently suggested for the Gakkel MOR[16], Papua New Guinea[17], Iceland[14], and parts of the Indian MOR[13]. However, the Southwestern Gondwana active margin was located far to the west[32,34], and hence, the SASWIR would have not yet interacted with a near-stationary

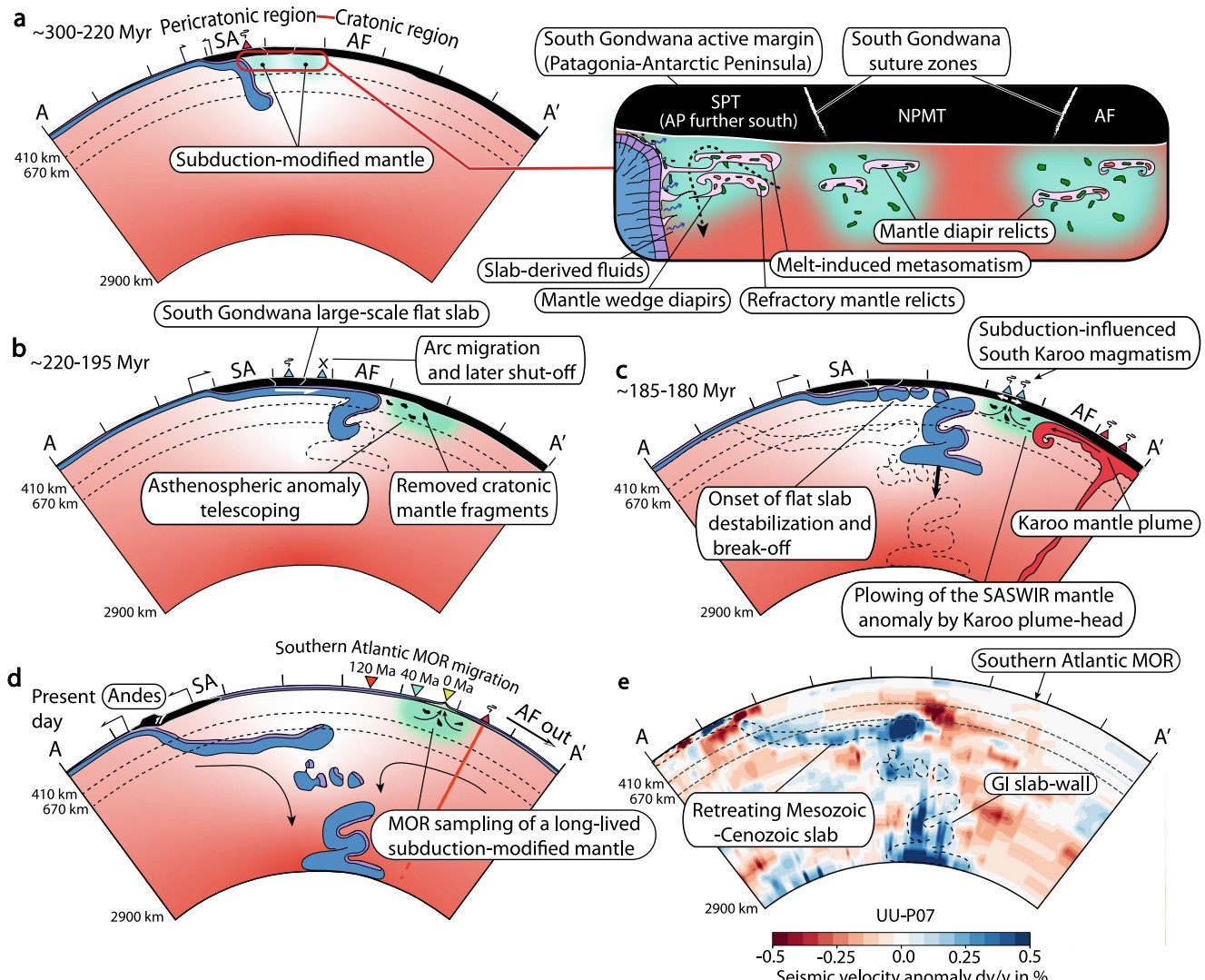

**Fig. 6 | Interpreted tomographic section and reconstruction of the Mesozoic-Cenozoic subduction system in Western South America illustrating the proposed asthenospheric anomaly telescoping process beneath the Southern Atlantic-Southwest Indian ridges. a–d** Reconstruction and interpretation of a **e**, cross-section from the UU-P07 global seismic tomography model[36] assuming an average whole-mantle slab sinking rate of 1.1 cm/yr[37]. Kinematics parameters for South Atlantic mid-ocean ridge and plate motion are from Müller et al.[32]. Neutrally to positively buoyant subduction-modified mantle anomalies in the inset sketch in **a**, are based on Richter et al.[16]. See cross-section location in Fig. 1. SA South American plate, AP Antarctic Peninsula, SPT South Patagonian terrane, NPMT North Patagonian terrane, AF African plate, and GI Georgia Islands slab.

geochemical mantle anomaly eventually formed beneath this area (Figs. 1, and 3d, f).

Our analysis of the mantle structure beneath the study area coupled with Mesozoic plate kinematic reconstructions[32–34] provide an alternative explanation for these issues. These reconstructions support a long-lived near-stationary subduction-modified mantle[12] located in front of the Early Jurassic subduction record of the Georgia Islands[34,38], which was previously interpreted as the remnant leading edge of an anomalously large flat slab[42] (Fig. 3d, e; Supplementary Fig. 3). In Fig. 6, we provide a 2-D reconstruction based on the interpretation of a cross-section from the UU-P07 P-wave global seismic tomography model[36] showing the proposed geodynamic evolution for the study area. Based on our reconstructions and the results from 2-D thermomechanical modeling, we suggest that the South Gondwana flat slab caused the frontal translation of the upper mantle at rates between ~6 and 5.08 cm/yr placing a subduction-modified mantle source underneath the Gondwana interior (Figs. 3e, 4, 5, and 6a, b). The South Gondwana flat slab would have initially caused an arc migration up to ~1000 km and then continued expanding amagmatically, at least

at upper crustal levels (Fig. 4). During this process, the subduction-modified mantle beneath the Patagonian-Antarctic Peninsula active margin and also that potentially beneath former suture zones[46,47] were jointly translated below the supercontinent and transported to more than 2280 km from the trench (Fig. 6a, b). After flat slab destabilization and break-off at ca. 185-180 Myr ago[42], the subduction-modified mantle remained relatively stationary in the mantle reference frame (Fig. 6c). This mantle source was partially sampled during the synextensional Karoo intraplate magmatism at ca. 186-176 Myr ago[12,25], possibly caused by Karoo mantle plume plowing (e.g.,[24]) (Fig. 6c). After the South Atlantic-Southwest Indian ocean opening, the subduction-modified mantle source was sampled again by the SASWIR since ca. 50-35 Myr ago as the ridge axis progressively overrode the near-stationary geochemical mantle anomaly (Figs. 3f and 6d, e). At the moment, the lack of enough off-axis geochemical data from the South Atlantic and Southwest Indian ocean floor does not allow a complete record of the geochemical influence of the SASWIR mantle anomaly since 50-35 Myr ago. Although of local character, the presence of 2.8 Ga continental zircon xenocrysts from 53-35 Myr MORBs from the

Shaka fracture zone[29] provides the only indirect evidence of early sampling of this mantle anomaly by the SASWIR considering that subduction is the most effective process to introduce sediments in the upper mantle (e.g.,[27]). Independent evidence of limited motion of mantle anomalies in the study area comes from two lines of evidence. First, the tomographically imaged delaminated blocks to the north of the SASWIR have remained neutrally buoyant in the upper-mantle and transition zone relatively close to their foundering site (~500 km) since ca. 120–130 Myr ago[52] (Supplementary Fig. 5). Second, hot-spots in the South Atlantic and Southwest Indian oceans indicate limited motion of a few millimeters per year over the last ca. 120 Myr ago[53].

Flat subduction is known to remove upper-plate cratonic lithospheric mantle during inland growth dumping lithospheric debris to the mantle wedge causing mantle source contamination (e.g.,[54]). Therefore, potential lithosphere erosion expected at the leading edge of the South Gondwana flat slab provides a viable explanation for the localized evidence of reworked lithospheric mantle in the magmatic source of the SASWIR[10,12] (Fig. 6b).

A comparison with the largest known flat slab events[49,50] indicates that the South Gondwana flat slab is likely the most extensive flat subduction episode documented so far (Fig. 4). The development and stability of this large-scale geodynamic process with a maximum potential size of ~2600 km, as well as the associated shifting of the sub-continental mantle, are supported by our 2-D thermomechanical model (Fig. 5). The forced migration of a subduction-influenced asthenosphere to form a near-stationary sub-lithospheric geochemical anomaly in the intraplate area differs from the documented flat slab-related mantle modification that imprints subduction geochemical anomalies in the lithospheric mantle within the moving upper plate[55]. Thus, we refer to this mechanism as 'asthenospheric anomaly telescoping'. This concept describes a previously unconsidered geodynamic consequence of flat subduction besides commonly acknowledged tectonomagmatic changes in the upper plate (e.g.,[49,50,56]). Flat slab subduction is a recurrent process with geological records back to the Mesoproterozoic[57] and possibly, a frequent subduction style in the Archean[58]. Therefore, it is reasonable to expect that the asthenospheric anomaly telescoping may have operated since the onset of plate tectonics, contributing to the development of upper mantle heterogeneities beneath continental interiors. This process would have been particularly effective for large-scale flat subduction that most likely terminates in slab break-off[59], precluding a trenchward suction of the subduction-modified mantle expected during the slab steepening. Our study provides a roadmap to test this hypothesis back to the late Paleozoic, which is as far as the mantle may record[34,38,41].

High vote counts interpreted as Late Triassic-Jurassic slabs in our vote map also depict a close spatial relationship to localized MORBs samples with ghost-arc signatures scattered along-strike the Central and Northern Atlantic MOR[7,14,19] (Supplementary Fig. 6). Thus, long-lived near-stationary geochemical anomalies associated with ancient subduction may provide an alternative to upper mantle convection for these anomalies as previously suggested by van Hinsbergen et al.[17]. Similarly, near-stationary subduction-modified sources in parts of the Indian MOR have been recently linked to Neoproterozoic subduction[13] or Mesozoic intra-Tethys Ocean subduction[16].

Finally, the restricted upper mantle flow in the study area since the last 185-180 Myr ago, indicated by the limited motion of the SASWIR subduction-modified mantle anomaly, has implications for recent models of South American plate motion and orogenesis that attribute a first-order role to strong westward mantle flow since Mesozoic[60]. Nevertheless, an accentuated mantle flow may be more recent (50-40 Myr ago) and mostly restricted to the north of the study area[61], favoring local preservation of the subduction-modified mantle beneath the SASWIR.

## Methods

### Kernel density map of MORB samples with ghost-arc geochemical signatures

We carried out a kernel density analysis of global MORB samples with backarc basin basalt-like geochemical signatures (i.e., ghost-arc geochemical signatures[16]) from the global compilation of Yang et al.[19] (Supplementary Fig. 1; Supplementary Data File 1). According to Yang et al.[19], the appropriate comparison for the signal that escapes the arc is not the arc front itself, which reflects the captured slab flux, but the backarc basin basalts that reflect a subduction signal emerging beyond the arc at a MOR. For this end, these authors used four geochemical ratios (Ba/Nb, Rb/Nb, Nb/U, and Ce/Pb) to build a 'backarc basin basalt filter' that provides a simple true or false test for subduction influence in samples from their global MORB dataset. To build the Kernel density map (Supplementary Fig. 1), we eliminated those samples from the dataset belonging to active backarc settings in Scotia, Mariana, and Bismarck Sea regions leaving 578 of the 1022 original samples. We applied the GQIS 3.22.6 "heat map" interpolation tool (available at https://www.qgis.org), defining a radius of influence of 2,9° (~320 km) and a parabolic Epanechnikov kernel function. This function minimizes variance, making it suitable for the continuous representation of clustered data within highly dispersed populations.

### Plate kinematic reconstructions and analysis of seismic tomography models

We carried out reconstructions of Gondwana in the Late Triassic-Early Jurassic using the Mesozoic plate kinematic model of Müller et al.[32] in the Gplates 2.0 software freely available at https://www.gplates.org. This model includes a recently developed approach[62] to reconstruct absolute plate motions back to 220 Myr ago in a mantle reference frame using a joint global inversion of multiple constraints, including hot spot location and associated trail data for the last 80 Myr, optimization of subduction zone migration behaviour considering global trench migration, and estimates of net lithospheric rotation, providing both paleo-latitudes and paleo-longitudes relative to the mantle. Also, we included Mesozoic plate kinematic reconstructions utilizing the mantle reference frames of Müller et al.[33], van der Meer et al.[34] and Torsvik and Cocks[35] (Supplementary Fig. 3). Müller et al.[33] extend the concept of 'tectonic rules-based mantle reference frame' introduced by Tetley et al.[62] by including the evaluation of continental velocities relative to the mantle as an additional criterion. The model of van der Meer et al.[34] is built on the paleomagnetic frame of Torsvik et al.[63] and constraints paleolongitude based on tomographic images of subducting slabs distributed in the mantle. Torsvik and Cocks[35] use a mantle reference frame that applies the plume generation zone method. This approach assumes long-term stability of large low-shear-velocity provinces and mantle plumes rising mainly at the edges of these lower mantle areas, allowing a longitudinal correction to reconstructed plates[35]. In our reconstructions, we have corrected the position of the Antarctic Peninsula that was located immediately to the west of the Austral Patagonian Andes (e.g.,[31,46]). The UU-P07 global tomography model and resolution tests for the Georgia Islands slab can be downloaded from https://www.atlas-of-the-underworld.org[38]. In Fig. 3d and Supplementary Fig. 3, we assume a constant and vertical slab sinking and considered minor lateral migration in the lower mantle after slab detachment following geological and geophysical studies[34,38,39,64] and recent numerical geodynamic models[65]. Also, we used an average whole mantle slab sinking rate of 1.1 cm/yr[37]. This value is between rates of 1 and 1.3 cm/yr that have demonstrated optimal correlations between the magmatic arc, tectonics, and sedimentary records and the mantle structure in Western South America for the Permian-Early Triassic[41], Late Triassic-Jurassic[40,42], and Cretaceous-Cenozoic[66,67], as well as in other ancient convergent margins[38,39,64]. Sensitivity analyses considering a range of whole mantle slab sinking rates can be found in Supplementary data in Gianni et al.[40].

## Vote map analysis

We built a positive wave speed vote map by employing the plotting tools from the submachine portal (https://www.earth.ox.ac.uk/ca.smachine/cgi/index.php) of Hosseini et al.[68] (Fig. 3c). We utilized 26 models (P and S-wave) with most of them differing in data selection and parametrization and regularization of the inversion (GyPSuM-S[69]; DETOX P2 and P3[70]; HMSL-P06 and S06[71]; PRI-P05 and -S05[72]; SPani-P and -S[73]; GAP-P4[74]; LLNL_G3Dv[75]; Hosseini2016[76]; SEISGLOB1[77]; MITP08[78]; UU-P07[36]; TX2019Slab-P and S[79]; S362ANI+M[80]; S20RTS[81]; S40RTS[82]; SAVANI[83]; SAW642ANb[84]; SEMUCB-WM1[85]; SEMum[86]; TX2011[87]; TX2015[88]; see model details in Supplementary Data Table 1) and implemented the standard deviation threshold following Shephard et al.[89].

## Determination of flat slab tip kinematics

For this analysis, we included the arc position from igneous rocks belonging to the magmatic arc from the geochronological dataset previously compiled by Navarrete et al.[42] (Supplementary Data file 3) with a starting arc position at ~500 km from the trench at ca. 220 Myr ago. Then, considering potential trench-Georgia Islands slab wall distances from the reconstructions including different plate kinematic models (2600 km[32], 2400 km[33], and 2280 km[34]) (Fig. 3d; Supplementary Fig. 3), and a time span of flat slab development of 35 Myr ago, we obtained flat slab tip migrations rates of 6 cm/yr, 5.42 cm/yr, and 5.08 cm/yr, respectively (Fig. 4). This analysis is justified because flat slabs commonly continue to expand laterally after arc-shut-off stages[50,54]. We are aware that a starting arc position at 500 km from the trench would indicate a shallowing slab before the peak flat subduction development. Reconstructions considering the mantle reference frame of Torsvik and Cocks[35] were excluded from this analysis because these yield results that are not compatible with the geological records of the western margin of Gondwana (see Results section).

## Numerical modeling method

We design a two-dimensional model to study the development of low-angle subduction. The conservation of mass, momentum, and energy equations are solved for an incompressible, viscoplastic fluid in a 2D Cartesian box using the finite element, particle-in-cell (PIC) code Underworld2[90–92]. Underworld2 follows a continuum mechanics approximation, which is widely used to describe geological and geophysical processes and solve the conservation equations of momentum, mass, and energy,

$$\nabla \cdot (\eta \nabla^s \mathbf{u}) + \nabla p = \rho \mathbf{g} \tag{1}$$

$$\nabla \cdot \mathbf{u} = 0 \tag{2}$$

$$\rho \, C_p \left( \frac{\partial T}{\partial t} + \mathbf{u}\nabla T \right) = \nabla \cdot (k\nabla T) + \rho f \tag{3}$$

where $u$ is the velocity, the $\nabla^s = 1/2 \left( \nabla + \nabla^T \right)$ is the symmetrized gradient, $\mathbf{n}$ the velocity, $T$ the temperature, $p$ the pressure, $\eta$ the dynamic viscosity, $\rho$ the density, $\mathbf{g}$ the gravitational acceleration vector, $C_p$ the isobaric heat capacity, $k$ the thermal conductivity, and $f$ a heat source term (accounting the decay of radioactive elements, adiabatic heating and shear heating).

We use nonlinear temperature and strain rate-dependent viscoplastic rheology. The viscous deformation of rocks is calculated using a power-law equation, with dislocation and diffusion creep determined through a generic relationship between stress and strain rate for each mechanism:

$$\dot{\varepsilon} = A(\sigma'/\mu)^n (b/d)^m exp\left( -\frac{E + PV}{RT} \right) \tag{4}$$

where $d$ is the average grain-size, $\sigma$ the deviatoric stress second invariant, $A$ the pre-exponential factor, $\mu$ the shear modulus, $b$ the length of the Burgers vector (i.e., lattice distortion), $n$ the stress exponent, $m$ the grain-size exponent, $E$ the activation energy, $V$ the activation volume, and $R$ the gas constant. Viscosity is limited in the model between $10^{19}$ and $10^{24}$ Pa s. Maximum strain rates in the model reach ~$10^{-14}$ s$^{-1}$, which produces a viscosity >$10^{19}$ Pa s for the rheology used. Supplementary Table 2 describes the dimensional values used in this study.

We have implemented a melt function to account for the thermal and mechanical effects of partial melting ($M$). This function, however, does not account for melt extraction processes so the code is restricted to partially melted regions in which the melt remains in its source. The mechanical effect associated with partial melting of the slab and surroundings is determined by a reduction of the viscosity of the lower crust within a melt range of 0.15 to 0.3. Melting is applied to existing viscous rheology, and is calculated as:

$$M_{int} = 1 + \left( \frac{M_f - L_f}{L_f - U_f} \right). \tag{5}$$

$$\eta_m = \eta \times (1 + M_{int} + \eta_f \times (1 - M_{int})), \tag{6}$$

where $\eta_m$ is the updated viscosity after material melts, $\eta$ is the viscous rheology, and $M_{int}$ is a normalized linear interpolation of the percentage of the melt fraction ($M_f$) between the upper ($U_f = 30\%$) and lower ($L_f = 15\%$) limit of the melt fraction range, and $\eta_f$ is the melt viscous softening factor that lower crust material undergoes once melted. When the melt fraction increases from 15% to 30%, the viscosity decreases by 2 orders of magnitude[93]. The melt fraction ($M_f$) is a function of the temperature and is calculated as:

$$T_{ss} = \frac{(T - (T_s + T_l) \times 0.5)}{(T_l - T_s)}, \tag{7}$$

$$M_f = 0.5 + T_{ss} + (T_{ss}^2 - 0.25) \times (0.4256 + 2.988 \times T_{ss}), \tag{8}$$

where $T_{ss}$ is the super-solidus, $T_s$ is the solidus temperature, and $T_l$ is the liquidus temperature. The solidus and liquidus for the slab are both temperature and pressure dependent and are parameterized by a polynomial relationship between temperature and pressure as

$$T_s = a_s + b_s P + c_s P^2, \tag{9}$$

$$T_l = a_l + b_l P + c_l P^2, \tag{10}$$

where $a$, $b$, and $c$ are constants and are defined in Supplementary Table 2.

Plastic failure is determined using a pressure-dependent Drucker–Prager yield criterion. The brittle properties of materials change as a result of a local strain accumulation, so that both cohesion and friction coefficient decrease linearly with strain. The yield stress linearly drops to a maximum of 20% of its initial value (or 2 MPa) when the accumulated strain reaches 0.5 for all materials in our simulations.

The initial model geometry is shown in Supplementary Fig. 4 and material properties and model parameters are presented in Supplementary Table 3. The model domain is two-dimensional, with a width of 5000 km and depth of 660 km. A uniform grid is used, with a

distribution of 1024 x 128 nodal points. The initial configuration is that of a homogeneous oceanic lithosphere with a 45° dipping weak zone within the mantle lithosphere at x = 1000 km. We follow Beaumont et al.[94] in choosing a base set of laboratory-derived rheology parameters. The continental upper and lower crust has a wet quartzite rheology[95]. The continental lower crust has a dry Maryland diabase rheology[96]. The oceanic crust and oceanic mantle lithosphere has the rheology of wet olivine[97]. The weak zone has a wet olivine rheology and low friction coefficient while the continental mantle lithospheric has a dry olivine rheology. Water content was added for the oceanic crust and weak zone rheology to favor the start of subduction[98]. A 30 km "sticky air" layer is included with a low viscosity ($10^{19}$ Pa s) and density (1 kg m$^{-3}$), which minimizes shear stresses at the surface and creates a pseudo free surface[99].

A constant temperature ($T = 0$ °C) is applied to the top boundary, with no heat flux across the side walls. The initial internal temperature distribution follows a geothermal gradient of 12 °C km$^{-1}$ until a temperature of 1300 °C is reached at the lithosphere-asthenosphere boundary (LAB) at a depth of 100 km. Below the base of the LAB, temperatures are calculated by linear interpolation between 1300 °C and 1573 °C. The model uses a free-slip condition on the right ($ux = 0$) and top ($uy = 0$) boundaries. The convergence velocity is applied on the left wall across the lithosphere in the first 200 km, which induces plate convergence at 6 cm/yr mainly driven by lower plate subduction consistent with the maximum possible rate derived from Fig. 4. Below 200 km, the velocity is 0 cm/yr. At the lower right corner of the model, we implemented an outflow equal to the inflow of the convergence velocity. Also, we track mantle flow during the model evolution in a wide area equivalent to the peri-cratonic region where subduction-modified mantle initially beneath the Patagonian-Antarctic Peninsula active margin[42–44] and intraplate suture zones[46,47] is expected to have translated frontally during flat slab development, as suggested in this study. The model reproduces the large-scale flat-subduction of about 35-40 Myr ago, compatible with flat slab kinematics derived from Fig. 4, and the accompanying inland shift of the sublithospheric mantle suggested in this study. In the numerical model, we consider the initial stage at 10 Myr, indicating the onset slab shallowing (Fig. 5).

## Data availability
All data needed to evaluate the conclusions in the paper are presented in this manuscript and the Supplementary information.

## Code availability
The Underworld2 code used for our 2-D numerical modelling is freely available at https://www.underworldcode.org/.

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

## Acknowledgements

G. M. G. and C. R. N. recognize the support given by CONICET and the funding given by the Universidad Nacional de la Patagonia San Juan Bosco (Grant number: CIUNPAT no. 1399). S. Z. and J. L. acknowledge the funding from the European Union's Horizon 2020 research and innovation programme under the Marie Sklodowska-Curie grant agreement No 777778. S.Z. acknowledges the funding of Project PID2020-113463RB-C32 funded by MCIN/AEI /10.13039/501100011033 and the funding of Generalitat de Catalunya via the 2021 SGR 01049.

## Author contributions

G.M.G. conceived the study, carried out the tomotectonic analyses, and wrote the draft of the manuscript. J.L. and S.Z. designed and carried out numerical subduction models. C.R.N. analyzed geochemical data and built Fig. 2. C.R.G. contributed to the geodynamic concepts in this study and built the kernel density map of supplementary fig. 1. All the authors contributed towards building figures, analyzing and interpreting the tomotectonic analyses, and experiments, and towards editing of the manuscript.

## Competing interests

The authors declare no competing interests.
