## [Peer Review File · Nature Communications]

REVIEWER COMMENTS

Reviewer #1 (Remarks to the Author):

Gianni et al. Review by Brendan Murphy

There are a plethora of geochemical and isotopic studies that have documented subtle ('ghost') arc signatures near ocean ridges and numerous explanations for these enigmatic signatures have been proposed. Both of these attributes are admirably described in the Introduction to Gianni et al., which also sets the stage for a further hypothesis; that they are related to the forced migration of subduction- modified mantle in response to the onset of flat-slab subduction.

As noted by several authors (see references 1,2,3 below), flat-slab subduction should be very common in the geological record, but its effects are rarely considered. What the authors present is geodynamic support for their very plausible concept. In my view, this manuscript provides a much better explanation for the time-space relationships than competing models.

I really enjoyed reading this manuscript, and I think it will be of great interest to the large cadre of researchers working at the interface of mantle geodynamics, tectonics, geochemistry and petrology. This particular study is based on "tomotectonic analysis and a new 2-D thermomechanical modeling". Although clearly beyond the scope of this article, I encourage the authors to compare the geochemistry of these regions before and after putative flat-slab events (taking into account the horizontal component of the slab tip migration time). There should be important "before and after" geochemical contrasts. In other words, this hypothesis is certainly more testable than its rivals. The rival hypotheses may not be mutually exclusive, but this one has a clear test.

As with many new ideas, this manuscript challenges us to think about long-term mantle evolution in a different way. I can see it forming an important component of graduate student seminars which evaluate the many ways in which the convective mantle has been contaminated directly or indirectly by subduction processes.

I have made numerous comments on an annotated version of the manuscript (attached). The vast majority of these comments are grammatical and/or editorial and can be handled with ease. Therefore I recommend publication after minor revisions. For the most part, the manuscript is clearly written. However, there were a few instances where I was puzzled with by some ambiguous statements. Most of the edits are suggestions for re-phrasing or clarification (many with an international readership in mind) and the authors should take care that my edits do not inadvertently change the meaning they intended.

References

1. Murphy, J.B., Oppliger, G.L., Brimhall Jr., G.H., and Hynes, A., 1998, Plume-modified orogeny: An example from the western United States: *Geology*, v. 26, p. 731–734, [http://dx.doi.org/10.1130/0091-7613\(1998\)026<0731:PMOAEF>2.3.CO;2](http://dx.doi.org/10.1130/0091-7613(1998)026<0731:PMOAEF>2.3.CO;2).
2. Fletcher, M., and Wyman, D.A., 2015, Mantle plume–subduction zone interactions over the past 60 Ma: *Lithos*, v. 233, p. 162–173, <http://dx.doi.org/10.1016/j.lithos.2015.06.026>.
3. Murphy, J.B., 2016. The role of the ancestral Yellowstone plume in the tectonic evolution of the western United States. *Geoscience Canada*, 43, 231-250.

Reviewer #2 (Remarks to the Author):

Dear editor,

Hereby I send you my review of Gianni et al's paper 'Ghost-arc geochemical anomaly at a spreading ridge caused by supersized flat subduction'

This paper starts with noting that 'ghost-arc' anomalies are one of the most enigmatic features in geodynamics, and I agree. These anomalies will likely dominate the geodynamic debate for the next decade, and there is a rapid growth in recognizing their importance and promise to understand mantle dynamics. So the subject of this paper is timely and very interesting.

The authors note a correlation between the SASWIR arc ghost and the Georgia Islands slab, which is also very interesting and on par with recent correlations on the Gakkel Ridge and New Guinea, as well as in the Central Atlantic. So there is a potential in this paper to add another correlation to the database, independently date this anomaly, and establish how long the mantle above GI has been stationary. If you can demonstrate that it did, this has major implications, for instance for explaining why South America is moving westwards (Schellart suggested it's the result of strong mantle flow, which would have moved this ghost for instance). But the paper does not really formulate a problem statement or hypothesis, but gives a model in the introduction instead.

Next the paper lengthily repeats a previous analysis of the authors that argues for a flat slab episode, whereby the ghost arc is used as a well-established and interpreted fact, rather than the exciting enigmatic feature that it is. The kinematic reconstructions are mixed with dynamic interpretations and modelling, for instance to support sinking rates, whereby it's unclear to me why that modelling is necessary. So much of the paper is a repetition of a point that was already made by the author team, rather than a new problem that is building on the previous work.

I think that the observations and the geological, kinematic, and tomographic work of the authors provides a promising starting point for a paper that could well become suited for this journal. But I think the current version lacks focus and organization and does not explain what the problem is, how that problem is methodically solved, and what the wider implications are. I hope the comments below help sharpening this.

Cheers,
Douwe van Hinsbergen
Utrecht, October 20, 2022

I. 55: I think it's relevant here to indicate that this concerns continental mantle lithosphere that broke in pieces during ancient rifting. Which gives you the same two options as 1 & 2: either rapid transport in a mantle that is flowing unrelated to plates, or a near-stationary mantle.

I. 83: migrate relative to the mantle,

I. 139-142: Don't present your model here yet. The reader hasn't seen any evidence yet, doesn't understand the setting or your model to explain the history. It is at this stage in the paper only confusing to come with major flat slab: I don't see how that really modifies the previous models. I also don't understand what this telescoping means. Instead of describing the model, describe the problem you are solving.

L. 144-151: you lost me here. I need to understand first what problem you are trying to solve, then see the evidence and reconstructions, and then I'll understand what you mean.

I. 152: What do you mean by 'tomotectonic'?

I. 154: I think it would be more useful if you use a kinematic model that is independent from the SASWIR anomaly to interpret when that anomaly formed. Don't use the anomaly as input in your reconstruction, you'll end up with circular reasoning.

I. 156: This flat slab is the only one of its kind so far proposed, and chances are, your readers are not immediately convinced about the validity of that model. Suppose your model is incorrect and needs modification, does that change anything for the SASWIR

anomaly or its history?

I. 157-166: This is becoming chaotic. The theme of ghost-arcs is not a well-established fact. You are the first to formally identify this as a ghost arc anomaly, so establish its existence first, summarize the arguments that this must be an arc remnant, note that it correlated with the GI slab, and then use an independent kinematic model to demonstrate that it can be explained by subduction at the Gondwana margin in the Triassic. And when you explain that kinematic model, you can notice that the slabs lie far inboard of the margin, and that you previously explained that by flat subduction, which poses the question how you can make a mantle wedge so far inboard, since slabs would have largely dehydrated by the time they reach the point where they dip down I suppose. Anyway, that's where your model comes in.

I. 165: not only an exceptional case, the only case by a long margin. The widest flat slab that I know of is a few hundred km, and India goes up to 800 km below Tibet. Your 2200 km are an order of magnitude larger.

I. 172: this is a repetition of what you said before.

I. 139-179 are out of place and are best deleted. Give the analysis first, because this is now coming across as arm-waving, and in a rather chaotic fashion. Which is not necessary, because you have identified something very interesting. So build up the story step by step.

L, 185: Please start with establishing that there is such a thing as a SASWIR ghost. Then go to interpreting its history. I see that you have missed the recent paper of Skublov et al 2022, (who found ancient zircons at fracture zones east of Bouvet. Fits the narrative of a subduction ghost.

I. 200: I am not a big fan of this term 'tomotectonic' that Sigloch uses. In the way her group is using it, they are making tectonic reconstructions by placing accreted geological records with a subduction history on top of anomalies that they speculate are accompanying slabs (but without a kinematic model to establish this, how do they know?), under the assumption that slabs do not laterally move through the mantle (which they do), and that their modern position reflects the position they always subducted in, as well as assuming mantle behavior and sinking rates. I don't know what you'd learn from models like that, particularly because geological constraints in their models are time and again violated. I know the work of your group, and you take a much more thorough, careful, and systematic approach in which you really study the geology, the tomography, and plate kinematics. I would not jump on the 'tomotectonic' bandwagon and use the buzzword, but simply explain to the reader how you plan to evaluate what that SASWIR anomaly is and how it formed.

L. 192: This reconstruction is not new, is it? Didn't you already write this down in a previous paper? If so, briefly explain that analysis and move on to interpreting the SASWIR ghost. Don't come with the whole analysis again.

L. 207: don't use numerical models. There are so many degrees of freedom in numerical models that you can make what you want. There's also models that have slabs race over the core-mantle boundary at plate tectonic rates, and Lijun Liu and a post-doc recently made one of slabs moving with 10 cm/yr criss-cross through the lower mantle. Stick with the kinematic reconstructions.

L. 207: We don't need numerical models to make kinematic correlations. We need kinematic correlations to inform numerical models.

I. 212-234: I don't understand why you are giving all of this information. Please start this section with a methodology: what is your problem, and how do you want to solve it. Why do you need to infer vertical sinking if you want to interpret the SASWIR ghost? All you need to do is to note that it is located above a lower mantle slab, as was noted for different ghosts before. And in analogy to those previous studies, you can then analyze what that

would mean for the longevity of this ghost. Or whatever problem you select.

L. 323: This is also your previous model, isn't it?

REVIEWERS' COMMENTS

Reviewer #2 (Remarks to the Author):

Dear editor,

I hereby send you my re-review of Gianni et al's paper 'Ghost-arc geochemical anomaly at a spreading ridge caused by supersized flat subduction', submitted to Nature Communications.

The authors have upgraded their manuscript using the previous round of review, and the paper is much easier to follow and better written. I have gone through the text once more and below append a list of suggestions. I think there are two main points that can be improved.

1) The paper provides an extensive set of original thoughts and arguments, which makes this an interesting contribution, but the 'hierarchy' in these arguments and thoughts can still be improved. I have given suggestions to this end in the detailed comments below, but particularly the role of the modeling in the analysis is not clear now. The way it is presented, a series of problems are solved in isolation. Why is there an arc signature at a ridge, how long can it be stable, what drives South America westward, how do you get a mantle wedge far below a continent, how do mantle wedges form and get transported by flat slabs, etc etc. It would be better to build this up in a hierarchy, whereby all but one of these problems need to be solved, step by step, to arrive at the answer of the biggest question. That'll increase the readability and impact of this paper.

2) The discussion is a succinct summary of the authors' train of thought, but it is mostly a repetition of the previous text. I would recommend weaving this story throughout the paper, and leave the discussion to only address the question: now that you have shown that this mantle wedge below the ridge formed in the Jurassic at the tip of a gigantic flat slab, what can you do with this information? That'll shorten the discussion and gives you room to expand on the 'so what' question.

I hope the authors find this final set of suggestions helpful to improve their paper.

Cheers,
Douwe

Gianni et al.: We have assessed these comments, which we think have led to a clearer organization of some parts of the manuscript. Detailed explanations to these and other issues are found in the answers to the comments below.

In this new revision we have modified the value of flat-slab migration rate in fig. 4 determined in the reconstructions using the mantle reference frame of van der Meer et al. (2010). We wrongly wrote it as 5.85 cm/yr while the correct value was 5.085 cm/yr (shown as 5,08). Please note that this doesn't modify our modeling results because we used the largest possible flat-slab migration rate (6 cm/yr).

I. 53: stationary in the mantle reference frame

Gianni et al.: Corrected.

I. 55: Option (iii) has indeed been proposed, but is not an explanation for ghost arc signatures, but for ancient cratonic signatures. You could remove it from the abstract.

Gianni et al.: We have modified the abstract following a comment below by this reviewer.

I. 57: what limitations? Can you be specific?

Gianni et al. : We have modified the abstract following a comment below by this reviewer.

I. 62: I still think you're burying the true signal below layers of jargon and models. Much of the abstract is now about previous work, I recommend that you focus immediately on the SASWIR anomaly. This is an enigmatic finding of arc signatures & old zircons that you interpret as a ghost arc. You notice that this arc is located above a slab in the lowermost mantle that is interpreted as a remnant of Triassic subduction. From this you infer that the ghost arc remnant may have remained stationary in the mantle since the Triassic. Moreover, the position of the GI slab lies far inboard of Pangea in the mantle reference frame, which has led you previously to infer that the GI slab formed during a period of flat slab subduction over a distance of 2200 km. From this you know that even for such wide flat slabs, arc signatures and sediments with zircons make it all the way to the slab bend. You support this by numerical modeling. Then add what we can do with this inference: why is this a high-impact finding? The implication may be (I guess) that even very wide flats slabs may still produce arc magmatism far inboard of a continent, consistent with findings in South China in the Mesozoic or so?

Gianni et al.: We have modified the abstract accordingly.

I. 81: 'Among the most widely acknowledged mantle source heterogeneities' is awkward phrasing. 99% of the community completely ignores these things, it's only in the last few years that some people have started to realize that these anomalies provide an opportunity to learn a few fundamental lessons about mantle dynamics, as you do in this paper. I'd rephrase to something like: In the last decades, a growing database of surprising geochemical signatures have suggested that the mantle below modern Mid-Ocean Ridges may contain ancient 'ghost' anomalies of plumes (refs), subduction (refs), or delaminated continental lithosphere (refs).

Gianni et al.: Sentence rephrased (see new lines 82-87).

L. 95, p4: linked to former subduction zones. Be as clear as you can.

Gianni et al.: Corrected (see new line 96).

I. 104: remains unclear

Gianni et al.: Corrected. (see new line 104).

I. 111: I'm curious how these modelers explain that the mantle is convecting opposite to plate motion. Otherwise you can't get the 'arc parcel' below a ridge...

Gianni et al.: Very interesting point!. We now include this comment in the introduction (see new lines 475-480).

I. 117: The key point is the mantle-stationary part. In the other model you also need to have long-lived preservation.

Gianni et al.: We removed the word long-lived (see new line116).

I. 118: What do you mean by 'stability'? I guess the long preservation potential? You can't explain the stability in the mantle reference frame this way.

Gianni et al.: Corrected (see new line 118)

I. 123: ascending into the mantle above the subducting slab by positive buoyancy in the arc and back-arc regions.

Gianni et al.: Corrected (see new line 123).

I. 128: That they're neutrally buoyant means that they float. Not that they remain mantle-stationary. That can only be explained if the upper mantle immediately below plates barely convects, as we pointed out in the van Hins 2020 GRL paper.

Gianni et al.: Corrected (see new lines 129-130).

I. 133: I'd say, or by other processes, such as plowing by continental edges (in collision or not doesn't really matter), or even by plume magmatism (e.g. Rojas-Agramonte et al., G-Cubed 2022).

Gianni et al.: Corrected (see new lines 129-134). We also added the suggested reference. Also, we added this possibility in the discussion section and in fig. 6c as a possible mechanism of SASWIR mantle anomaly sampling at 185 MYR ago by the impingement of the Karro mantle plume. Thanks for the helpful reference!

I. 154: what is 'mantle detritus'?

Gianni et al.: The reviewer is right. We have changed the word detritus by fragments (see new line 154)

I. 167: what is unclear about the tomography? The models are not so important, they simply should obey and explain observations.

Gianni et al.: Corrected (see new lines 167-168).

I. 169: subduction-modified

Gianni et al.: Corrected (see new line 169).

I. 169-174: This is quite a lot of discussion and you may lose your audience here. Come to the point of your analysis instead.

Gianni et al.: We have thought a lot about your point. But we decided to keep this discussion because here is where we tell the reader what are the weaknesses of the previous model of upper mantle

convection as the favored interpretation for the origin of the SASWIR mantle anomaly. This allows us to introduce in the following paragraph what we will do to shed light into this issue.

I. 174: 'this model': I have lost track of which model you referred to by now.

Gianni et al.: We have corrected this sentence to clarify to which model we are referring (see new lines 172-179).

I. 179; which issue?

Gianni et al.: We have corrected this sentence to clarify that we refer to the origin of the SASWIR subduction-modified mantle anomaly (see new line 180).

I. 186: You're doing everything at the same time here. Build it up layer by layer: you first evaluate whether the anomaly is associated with a slab in the mantle (like Richter, or us in Papua etc). Then you evaluate which subduction zone made that slab, and when. Then you infer how long the anomaly existed, what caused it, etc.

Gianni et al.: The reviewer is right. We have modified this part accordingly (see new lines 180-204).

I. 186-192: Delete the last sentence 'Based on...SASWIR': show the analysis first before you push for the conclusion.

Gianni et al.: We understand the reviewer's point. Nevertheless, we must leave these last sentences because the journal requires the last paragraph of the introduction to be a summary of our findings.

L

. 244-286: 'Also...SASWIR': this section should come later in the paper. First describe the anomaly, then link to tomography, then link the slab to a paleo-subduction zone, then independently confirm with this Karoo link in the Jurassic that a subduction-enriched wedge may already have been present here in the Jurassic.

Gianni et al.: We have discussed with the rest of the coauthors about moving this part, but we decided to keep it where it is. We think that demonstrating since the beginning of the results section that the SASWIR mantle anomaly is a Jurassic feature paves the way towards the tomographic analysis of Jurassic subduction mantle records in the following part of this section.

I. 259: For what did you use UUP07, and for what the vote maps?

Gianni et al.: We now explain this point (see new lines 277).

I. 277: 2150 km and deeper

Gianni et al.: Corrected (see new line 295).

I. 280: at analyzed depths? I don't understand what you mean.

Gianni et al.: This is because assuming average whole-mantle slab sinking rates of 1-1,2 cm/yr at a depth of 2150 km the GI records Jurassic subduction, but at the base the GI records subduction until Permian times (van der Meer et al. 2018) (see new line 300).

I. 287 and onwards: this is a next paragraph, in which you add time to the picture.

Gianni et al.: Corrected (see new line 307).

I. 292: this should be ref 36.

Gianni et al.: Corrected to 38 (former ref. 36) (see new line 312).

I. 302: Does the reconstructed inboard magmatic arc end up above the position of the GI slab?

Gianni et al.: No, the magmatic arc shutted down at ~1200 km from the trench, but the slab tip must have continued migrating inland as occurred in the Laramide flat-slab. This is explained in more detail in the discussion section where we described our model of Fig. 6.

I.305: South Africa: is this the formation of the Cape Fold-belt?

Gianni et al.: Yes, but only the last shortening stage. As reviewed by Navarrete et al. (2019), the Late Triassic flat slab model explains a series of compressional reactivations of permian fold and thrust belts (within them, the Cape fold belt), positive basin inversion of former extensional basins in Patagonia, and ductile shearing in the active Patagonian-Antarctic Peninsula active margin (Riley et al., 2020; Suárez et al., 2020).

I. 311: what kind of alternative reference frames? You need a mantle frame, otherwise you don't have paleolongitudinal control. And for the Jurassic, all you have is van der Meer et al., 2010, Torsvik and Cocks 2017 (you can download the reconstruction from the CEED website), and the recent 'tectonic rules' frame of Muller et al., 2022. No other frames go back far enough as far as I know.

Gianni et al.: The reviewer is totally right. Now, we only use mantle reference frames as suggested and removed our analysis including the model by Müller et al. (2016). Please note that the plate kinematic model of Muller et al. (2019) originally applied in this study (Fig. 3d), uses the mantle reference frame of Tetley et al. (2019). The latter authors presented a method applying a joint global inversion to evaluate the contribution of multiple time-dependent absolute plate motion constraints including fit to age-progressive hotspot tracks, optimizing subduction zone migration behaviors, and minimizing rates of net lithospheric rotation. This method provides both paleo-latitudes and paleo-longitudes relative to the mantle. This approach has been extended in Muller et al. (2022) by including evaluation of continental velocities relative to the mantle as an additional criterion and extending the tectonic rules-based approach proposed by Tetley et al. (2019) to the last billion years. Now, we use the mantle reference frame of Muller et al. (2022) as suggested by the reviewer. Also, we included the plate kinematic model of Torsvik & Cocks (2016) that applies the PGZ method for their mantle reference frame as suggested by the reviewer (See new fig. 3 and new supplementary fig. 3). The spatio-temporal relationship between the mantle and surface geological records of flat subduction is supported by plate kinematic reconstructions applying the different mantle reference frames (Muller et al. 2019 and 2022, van der Meer et al., 2010, Fig. 3d, Supplementary Fig. 3a,b). The only exception is in the reconstruction with the mantle reference frame of Torsvik and Cocks (2016). This presents similar results to the rest of the reconstructions at 220-210 Myr ago but differ since 195 Myr ago, depicting major offsets between the mantle slabs (mainly in the central and northern Pangea margin) and the reconstructed trench (Supplementary Fig. 3c). This observation would indicate intra-oceanic subduction offboard the Southwestern margin of Gondwana at this time, which is not

compatible with the geological records of an Early Jurassic continental arc and back arc basins in western South America. This could be possibly caused by limitations (see Flaments et al., 2022) in the underlying assumptions in the plume generation method used in the reconstructions by Torsvik and Cocks (2016). Thus, we consider Müller et al (2019, 2022) and van der Meer et al. (2010) as the most consistent with the geological records in the active margin of southwestern Gondwana.

L. 324-328: Rather than arguing for this conclusion, explain to the reader what the problem is that we now need to solve if we accept that the SASWIR anomaly is linked to the GI slab, which is linked to Jurassic flat slab subduction.

Gianni et al.: The reviewer is right. We have modified this part as suggested (see new lines 351-358).

I. 331: Flat slabs cannot be caused by slab retreat. They are either advancing, or retreating slower than upper plates, but retreat always makes flat slabs smaller than if there had not been no retreat.

Gianni et al.: 'the reviewer is right. We now indicate that it is a 'forced' trench retreat as suggested by Sheppers et al. (see new lines 370-371).

I. 338: depending on the mantle frame rather than plate model. Also: you use the frames of Muller here, but how did they constrain the paleolongitude before the Cretaceous? I don't think they had paleolongitudinal control. They arguably do in their 'tectonic rules' frame (2022 Solid Earth), so that one would make more sense to use. And the plume-LLSVP-fitted frame of Torsvik & Cocks 2017.

Gianni et al.: The reviewer is right. We have addressed this comment by including the mantle reference frames of Muller et al. (2022) and Torsvik and Cocks (2016) in addition to that of van der Meer et al. (2010) and Müller et al. (2019) (see new Fig. 3a, supplementary fig. 3, and fig. 4). Please note that Muller et al. (2019) shown in fig. 3, already includes the mantle reference frame of Tetley et al. (2019), the first to suggest the tectonic rules-based mantle reference frame. We opted to show reconstructions with the mantle reference frame of Tetley et al. (2019) in fig. 3d because it provides the maximum possible paleo-trench-Georgia islands slab distance from which we derived the kinematics included in our 2-D numerical modeling. Also, note that with the exception of the reconstructions with the mantle reference frame of Torsvik and Cocks (2016), which yield results not compatible with the geological records at 195 Ma (see comment above), the other three reference frames yield very similar paleo-trench-slab distances (22800-2600 km) (Fig. 3d, Supplementary fig. 3, fig. 4).

I. 345: I can see this line of reasoning, but it would help the reader if you first explain what the problem is that you try to solve with this analysis. What are you trying to learn? I mean: once you establish a link between the GI slab and the SASWIR anomaly, the problem to explain the anomaly is solved. You don't need to model flat slabs for that anymore. So the step to the flat slabs is a next layer: you are now using the SASWIR anomaly to learn something about the geochemical traces of flat slab events or so? Please explain what you want to solve with this analysis.

Gianni et al.: The reviewer is right. We have added a sentence at the beginning of this paragraph to highlight to the readers that the location of the Georgia Islands slab and the SASWIR mantle anomaly both far from the reconstructed margins, is telling us something important about how this anomaly was formed so far from the active margin (see new lines 360-365).

I. 375: Depending on the mantle frame, not the plate kinematic model

Gianni et al.: Corrected (see new line 375).

I. 393: A key aspect for what?

Gianni et al.: Corrected (see new line 428).

I. 395: Are you testing the feasibility? Suppose that you are not able to model this flat slab, would you then conclude that the SASWIR anomaly is not related to subduction? I don't think so, right? Aren't you rather using the correlations you made to infer the conditions under which the flat subduction occurred? Wouldn't that be more interesting? I'm not sure what it is you want to learn through the modeling.

Gianni et al.: The reviewer is right. We have modified these sentences accordingly (see new lines 432-435).

L. 401: I think it's the other way around. You reconstructed THAT the geochemical anomaly was transported far inboard, and still remains vertically above the associated slab. So now you have to 'train' your model to have that as a result, and you can learn about the dynamics of very large flat slabs. Or so.

Gianni et al.: We agree with the reviewer. This sentence has been modified accordingly (see new lines 432-440).

I. 417: As presented, you just add a numerical 'cartoon' of a model you already interpreted before the modeling started. I think you can do more with the modeling, to learn about conditions you cannot reconstruct (rheology, whatever). You can leave it as is, but it doesn't add much.

Gianni et al.: We are using the 2-D numerical model to understand if the flat-slab phenomenon of more than 2000 km is thermodynamically viable under the obvious limitations of a numerical model. We are interested in studying the conditions under which a flat-slab of these characteristics is viable. We have done extensive testing to evaluate rheological and thermal parameters to constrain those variables within previously reported value frames. It is clear that it is interesting to carry out a meticulous analysis of the behavior of the flat-slab by modifying these variables (which we did and we are working to publish the results) and gain deeper insights into the causes of large-scale flat subduction, but it seemed to us that it did not fall within the scope of this work.

I. 437: Would that give the same signature as an arc/mantle wedge?

Gianni et al.: Yes, if it was metasomatized by slab fluids. We added this in the sentence to clarify this point (see new lines 481-483).

I. 446: detritus is something I'd associate with a sandstone. Wouldn't 'fragments' be a better word?

Gianni et al.: The reviewer is right. This is now corrected (see new lines 492).

I. 457: well, basically, the explanation for the anomaly is the same in this scenario and in the previous ones, but the prediction for mantle flow is very different. In the former explanations, the uppermost mantle would flow opposite to plate motion. What would drive that?

Gianni et al.: Now we indicate that this model implies an eastward upper mantle flow of subduction-influenced asthenosphere that opposes continental plate motion in the study area, which is not easy to explain from a geodynamic point of view (see new lines 475-480).

I. 464: Explain what the problem is: the Gondwana margin was located far to the west of the SASWIR.

Gianni et al.: Corrected (see new lines 507-512).

I. 494: 'inland' is a bit strange here. 'Below the supercontinent'? Also: and transported to more than 2200 km etc

Gianni et al.: Corrected (see new lines 542-545).

I. 561: this term 'asthenospheric anomaly telescoping' is not very helpful to explain what's going on. I don't understand what it means, or why it's important. You just bulldozed mantle wedge eastward with a flat slab.

Gianni et al.: This concept means that a relict subduction-related mantle must not necessarily lie beneath a reconstructed active margin as suggested previously. The telescoping of asthenospheric anomalies suggest that relict subduction-influenced mantle, whether beneath the active margin or beneath intraplate areas (e.g., suture zones), can be jointly accumulated beneath intraplate interiors far from the plate margin creating long-term mantle heterogeneities.

I. 575: why would flat slab subduction result in slab break-off? I'd argue the opposite. You first need to roll that entire slab back, and then it'll break off. Or not (e.g., Boschman et al., G-Cubed 2018).

Gianni et al.: Not always. Some flat-slabs present arc migration patterns that depart from typical trenchward magmatic migration expected during slab steepening and roll back. This is a nice study discussing this issue: Dai, L., Wang, L., Lou, D., Li, Z. H., Dong, H., Ma, F., ... & Yu, S. (2020). Slab rollback versus delamination: Contrasting fates of flat-slab subduction and implications for South China evolution in the Mesozoic. *Journal of Geophysical Research: Solid Earth*, 125(4), e2019JB019164.

I. 543-581: This section contains a series of more or less loose thoughts, but how is this linked to the SASWIR anomalies? What do we really learn from your correlations?

Gianni et al.: We understand this point of view. Nevertheless, we have decided to keep this section as it is. From our point of view, this section summarizes three important aspects that we have learned from this study: 1) the South Gondwana flat slab is likely the largest documented flat subduction event, 2) we note that inland bulldozing of asthenospheric material to create long-lasting near-stationary subduction-related mantle anomaly is an unconsidered geodynamic consequence of flat slabs, and hence, we suggest a name for this geodynamic process, and 3) as flat subduction is common in active subduction settings, and likely was more common in the past, this must have been an important contributor to the development of upper mantle heterogeneities beneath continental interiors.

l. 591: well, this is not a speculation, but a deduction. If you have a neutrally buoyant wedge in the mantle, and it's not moving, then the mantle is not moving. You can't have one without the other.

Gianni et al.: Corrected (see new line 648).

l. 604: this is the same point as you made before.

Gianni et al.: We have modified this sentence (see new lines 655-661).

We are deeply thankful to Dr. J. J. Van Hinsbergen for his thorough, constructive, and kind revision of our manuscript.

Guido M. Gianni, Jeremías Likerman, César R. Navarrete, Conrado R. Gianni, and Sergio Zlotnik

REVIEWERS' COMMENTS

Reviewer #1 (Remarks to the Author):

The authors have done a most conscientious job in accommodating the concerns I raised in my previous review. I recommend acceptance.

Reviewer #2 (Remarks to the Author):

The authors have upgraded their manuscript using the previous round of review, and the paper is much easier to follow and better written. I have gone through the text once more and below append a list of suggestions. I think there are two main points that can be improved.

1) The paper provides an extensive set of original thoughts and arguments, which makes this an interesting contribution, but the 'hierarchy' in these arguments and thoughts can still be improved. I have given suggestions to this end in the detailed comments below, but particularly the role of the modeling in the analysis is not clear now. The way it is presented, a series of problems are solved in isolation. Why is there an arc signature at a ridge, how long can it be stable, what drives South America westward, how do you get a mantle wedge far below a continent, how do mantle wedges form and get transported by flat slabs, etc etc. It would be better to build this up in a hierarchy, whereby all but one of these problems need to be solved, step by step, to arrive at the answer of the biggest question. That'll increase the readability and impact of this paper.

2) The discussion is a succinct summary of the authors' train of thought, but it is mostly a repetition of the previous text. I would recommend weaving this story throughout the paper, and leave the discussion to only address the question: now that you have shown that this mantle wedge below the ridge formed in the Jurassic at the tip of a gigantic flat slab, what can you do with this information? That'll shorten the discussion and gives you room to expand on the 'so what' question.

I hope the authors find this final set of suggestions helpful to improve their paper.

Cheers,
Douwe

I. 53: stationary in the mantle reference frame

I. 55: Option (iii) has indeed be proposed, but is not an explanation for ghost arc signatures, but for ancient cratonic signatures. You could remove from the abstract.

I. 57: what limitations? Can you be specific?

I. 62: I still think you're burying the true signal below layers of jargon and models. Much of the abstract is now about previous work, I recommend that you focus immediately on the SASWIR anomaly. This is an enigmatic finding of arc signatures & old zircons that you interpret as a ghost arc. You notice that this arc is located above a slab in the lowermost mantle that is interpreted as a remnant of Triassic subduction. From this you infer that the ghost arc remnant may have remained stationary in the mantle since the Triassic. Moreover, the position of the GI slab lies far inboard of Pangea in the mantle reference frame, which has led you previously to infer that the GI slab formed during a period of flat slab subduction over a distance of 2200 km. From this you now that even for such wide flat slabs, arc signatures and sediments with zircons make it all the way to the slab bend. You support this by numerical modeling. Then add what we can do with this inference: why is this a high-impact finding? The implication may be (I guess) that even very wide flats slabs may still produce arc magmatism far inboard of a continent, consistent with findings in South China in the Mesozoic or so?

I. 81: 'Among the most widely acknowledged mantle source heterogeneities' is awkward phrasing. 99% of the community completely ignores these things, it's only in the last few years that some

people have started to realize that these anomalies provide an opportunity to learn a few fundamental lessons about mantle dynamics, as you do in this paper. I'd rephrase to something like: In the last decades, a growing database of surprising geochemical signatures have suggested that the mantle below modern Mid-Ocean Ridges may contain ancient 'ghost' anomalies of plumes (refs), subduction (refs), or delaminated continental lithosphere (refs).

L. 95, p4: linked to former subduction zones. Be as clear as you can.

I. 104: remains unclear

I. 111: I'm curious how these modelers explain that the mantle is convecting opposite to plate motion. Otherwise you can't get the 'arc parcel' below a ridge...

I. 117: The key point is the mantle-stationary part. In the other model you also need to have long-lived preservation.

I. 118: What do you mean by 'stability'? I guess the long preservation potential? You can't explain the stability in the mantle reference frame this way.

I. 123: ascending into the mantle above the subducting slab by positive buoyancy in the arc and back-arc regions.

I. 128: That they're neutrally buoyant means that they float. Not that they remain mantle-stationary. That can only be explained if the upper mantle immediately below plates barely convects, as we pointed out in the van Hins 2020 GRL paper.

I. 133: I'd say, or by other processes, such as plowing by continental edges (in collision or not doesn't really matter), or even by plume magmatism (e.g. Rojas-Agramonte et al., G-Cubed 2022).

I. 154: what is 'mantle detritus'?

I. 167: what is unclear about the tomography? The models are not so important, they simply should obey and explain observations.

I. 169: subduction-modified

I. 169-174: This is quite a lot of discussion and you may lose your audience here. Come to the point of your analysis instead.

I. 174: 'this model': I have lost track of which model you referred to by now.

I. 179; which issue?

I. 186: You're doing everything at the same time here. Build it up layer by layer: you first evaluate whether the anomaly is associated with a slab in the mantle (like Richter, or us in Papua etc). Then you evaluate which subduction zone made that slab, and when. Then you infer how long the anomaly existed, what caused it, etc.

I. 186-192: Delete the last sentence 'Based on...SASWIR': show the analysis first before you push for the conclusion.

I. 244-286: 'Also...SASWIR': this section should come later in the paper. First describe the anomaly, then link to tomography, then link the slab to a paleo-subduction zone, then independently confirm with this Karoo link in the Jurassic that a subduction-enriched wedge may already have been present here in the Jurassic.

I. 259: For what did you use UUP07, and for what the vote maps?

I. 277: 2150 km and deeper

- I. 280: at analyzed depths? I don't understand what you mean.
- I. 287 and onwards: this is a next paragraph, in which you add time to the picture.
- I. 292: this should be ref 36.
- I. 302: Does the reconstructed inboard magmatic arc end up above the position of the GI slab?
- I.305: South Africa: is this the formation of the Cape Foldbelt?
- I. 311: what kind of alternative reference frames? You need a mantle frame, otherwise you don't have paleolongitudinal control. And for the Jurassic, all you have is van der Meer et al., 2010, Torsvik and Cocks 2017 (you can download the reconstruction from the CEED website), and the recent 'tectonic rules' frame of Muller et al., 2022. No other frames go back far enough as far as I know.
- L. 324-328: Rather than arguing for this conclusion, explain to the reader what the problem is that we now need to solve if we accept that the SASWIR anomaly is linked to the GI slab, which is linked to Jurassic flat slab subduction.
- I. 331: Flat slabs cannot be caused by slab retreat. They are either advancing, or retreating slower than upper plates, but retreat always makes flat slabs smaller than if there had not been no retreat.
- I. 338: depending on the mantle frame rather than plate model. Also: you use the frames of Muller here, but how did they constrain the paleolongitude before the Cretaceous? I don't think they had paleolongitudinal control. They arguably do in their 'tectonic rules' frame (2022 Solid Earth), so that one would make more sense to use. And the plume-LLSVP-fitted frame of Torsvik & Cocks 2017.
- I. 345: I can see this line of reasoning, but it would help the reader if you first explain what the problem is that you try to solve with this analysis. What are you trying to learn? I mean: once you establish a link between the GI slab and the SASWIR anomaly, the problem to explain the anomaly is solved. You don't need to model flat slabs for that anymore. So the step to the flat slabs is a next layer: you are now using the SASWIR anomaly to learn something about the geochemical traces of flat slab events or so? Please explain what you want to solve with this analysis.
- I. 375: Depending on the mantle frame, not the plate kinematic model
- I. 393: A key aspect for what?
- I. 395: Are you testing the feasibility? Suppose that you are not able to model this flat slab, would you then conclude that the SASWIR anomaly is not related to subduction? I don't think so, right? Aren't you rather using the correlations you made to infer the conditions under which the flat subduction occurred? Wouldn't that be more interesting? I'm not sure what it is you want to learn through the modeling.
- L. 401: I think it's the other way around. You reconstructed THAT the geochemical anomaly was transported far inboard, and still remains vertically above the associated slab. So now you have to 'train' your model to have that as a result, and you can learn about the dynamics of very large flat slabs. Or so.
- I. 417: As presented, you just add a numerical 'cartoon' of a model you already interpreted before the modeling started. I think you can do more with the modeling, to learn about conditions you cannot reconstruct (rheology, whatever). You can leave it as is, but it doesn't add much.
- I. 437: Would that give the same signature as an arc/mantle wedge?
- I. 446: detritus is something I'd associate with a sandstone. Wouldn't 'fragments' be a better word?
- I. 457: well, basically, the explanation for the anomaly is the same in this scenario and in the previous ones, but the prediction for mantle flow is very different. In the former explanations, the uppermost

mantle would flow opposite to plate motion. What would drive that?

I. 464: Explain what the problem is: the Gondwana margin was located far to the west of the SASWIR.

I. 494: 'inland' is a bit strange here. 'Below the supercontinent'? Also: and transported to more than 2200 km etc

I. 561: this term 'asthenospheric anomaly telescoping' is not very helpful to explain what's going on. I don't understand what it means, or why it's important. You just bulldozed mantle wedge eastward with a flat slab.

I. 575: why would flat slab subduction result in slab break-off? I'd argue the opposite. You first need to roll that entire slab back, and then it'll break off. Or not (e.g., Boschman et al., G-Cubed 2018).

I. 543-581: This section contains a series of more or less loose thoughts, but how is this linked to the SASWIR anomalies? What do we really learn from your correlations?

I. 591: well, this is not a speculation, but deduction. If you have a neutrally buoyant wedge in the mantle, and it's not moving, then the mantle is not moving. You can't have one without the other.

I. 604: this is the same point as you made before.

REVIEWERS' COMMENTS

Reviewer #2 (Remarks to the Author):

Dear editor,

I hereby send you my re-review of Gianni et al's paper 'Ghost-arc geochemical anomaly at a spreading ridge caused by supersized flat subduction', submitted to Nature Communications.

The authors have upgraded their manuscript using the previous round of review, and the paper is much easier to follow and better written. I have gone through the text once more and below append a list of suggestions. I think there are two main points that can be improved.

1) The paper provides an extensive set of original thoughts and arguments, which makes this an interesting contribution, but the 'hierarchy' in these arguments and thoughts can still be improved. I have given suggestions to this end in the detailed comments below, but particularly the role of the modeling in the analysis is not clear now. The way it is presented, a series of problems are solved in isolation. Why is there an arc signature at a ridge, how long can it be stable, what drives South America westward, how do you get a mantle wedge far below a continent, how do mantle wedges form and get transported by flat slabs, etc etc. It would be better to build this up in a hierarchy, whereby all but one of these problems need to be solved, step by step, to arrive at the answer of the biggest question. That'll increase the readability and impact of this paper.

2) The discussion is a succinct summary of the authors' train of thought, but it is mostly a repetition of the previous text. I would recommend weaving this story throughout the paper, and leave the discussion to only address the question: now that you have shown that this mantle wedge below the ridge formed in the Jurassic at the tip of a gigantic flat slab, what can you do with this information? That'll shorten the discussion and gives you room to expand on the 'so what' question.

I hope the authors find this final set of suggestions helpful to improve their paper.

Cheers,
Douwe

Gianni et al.: We have assessed these comments, which we think have led to a clearer organization of some parts of the manuscript. Detailed explanations to these and other issues are found in the answers to the comments below.

In this new revision we have modified the value of flat-slab migration rate in fig. 4 determined in the reconstructions using the mantle reference frame of van der Meer et al. (2010). We wrongly wrote it as 5.85 cm/yr while the correct value was 5.085 cm/yr (shown as 5,08). Please note that this doesn't modify our modeling results because we used the largest possible flat-slab migration rate (6 cm/yr).

I. 53: stationary in the mantle reference frame

Gianni et al.: Corrected.

I. 55: Option (iii) has indeed been proposed, but is not an explanation for ghost arc signatures, but for ancient cratonic signatures. You could remove it from the abstract.

Gianni et al.: We have modified the abstract following a comment below by this reviewer.

I. 57: what limitations? Can you be specific?

Gianni et al. : We have modified the abstract following a comment below by this reviewer.

I. 62: I still think you're burying the true signal below layers of jargon and models. Much of the abstract is now about previous work, I recommend that you focus immediately on the SASWIR anomaly. This is an enigmatic finding of arc signatures & old zircons that you interpret as a ghost arc. You notice that this arc is located above a slab in the lowermost mantle that is interpreted as a remnant of Triassic subduction. From this you infer that the ghost arc remnant may have remained stationary in the mantle since the Triassic. Moreover, the position of the GI slab lies far inboard of Pangea in the mantle reference frame, which has led you previously to infer that the GI slab formed during a period of flat slab subduction over a distance of 2200 km. From this you know that even for such wide flat slabs, arc signatures and sediments with zircons make it all the way to the slab bend. You support this by numerical modeling. Then add what we can do with this inference: why is this a high-impact finding? The implication may be (I guess) that even very wide flats slabs may still produce arc magmatism far inboard of a continent, consistent with findings in South China in the Mesozoic or so?

Gianni et al.: We have modified the abstract accordingly.

I. 81: 'Among the most widely acknowledged mantle source heterogeneities' is awkward phrasing. 99% of the community completely ignores these things, it's only in the last few years that some people have started to realize that these anomalies provide an opportunity to learn a few fundamental lessons about mantle dynamics, as you do in this paper. I'd rephrase to something like: In the last decades, a growing database of surprising geochemical signatures have suggested that the mantle below modern Mid-Ocean Ridges may contain ancient 'ghost' anomalies of plumes (refs), subduction (refs), or delaminated continental lithosphere (refs).

Gianni et al.: Sentence rephrased (see new lines 82-87).

L. 95, p4: linked to former subduction zones. Be as clear as you can.

Gianni et al.: Corrected (see new line 96).

I. 104: remains unclear

Gianni et al.: Corrected. (see new line 104).

I. 111: I'm curious how these modelers explain that the mantle is convecting opposite to plate motion. Otherwise you can't get the 'arc parcel' below a ridge...

Gianni et al.: Very interesting point!. We now include this comment in the introduction (see new lines 475-480).

I. 117: The key point is the mantle-stationary part. In the other model you also need to have long-lived preservation.

Gianni et al.: We removed the word long-lived (see new line116).

I. 118: What do you mean by 'stability'? I guess the long preservation potential? You can't explain the stability in the mantle reference frame this way.

Gianni et al.: Corrected (see new line 118)

I. 123: ascending into the mantle above the subducting slab by positive buoyancy in the arc and back-arc regions.

Gianni et al.: Corrected (see new line 123).

I. 128: That they're neutrally buoyant means that they float. Not that they remain mantle-stationary. That can only be explained if the upper mantle immediately below plates barely convects, as we pointed out in the van Hins 2020 GRL paper.

Gianni et al.: Corrected (see new lines 129-130).

I. 133: I'd say, or by other processes, such as plowing by continental edges (in collision or not doesn't really matter), or even by plume magmatism (e.g. Rojas-Agramonte et al., G-Cubed 2022).

Gianni et al.: Corrected (see new lines 129-134). We also added the suggested reference. Also, we added this possibility in the discussion section and in fig. 6c as a possible mechanism of SASWIR mantle anomaly sampling at 185 MYR ago by the impingement of the Karro mantle plume. Thanks for the helpful reference!

I. 154: what is 'mantle detritus'?

Gianni et al.: The reviewer is right. We have changed the word detritus by fragments (see new line 154)

I. 167: what is unclear about the tomography? The models are not so important, they simply should obey and explain observations.

Gianni et al.: Corrected (see new lines 167-168).

I. 169: subduction-modified

Gianni et al.: Corrected (see new line 169).

I. 169-174: This is quite a lot of discussion and you may lose your audience here. Come to the point of your analysis instead.

Gianni et al.: We have thought a lot about your point. But we decided to keep this discussion because here is where we tell the reader what are the weaknesses of the previous model of upper mantle

convection as the favored interpretation for the origin of the SASWIR mantle anomaly. This allows us to introduce in the following paragraph what we will do to shed light into this issue.

I. 174: 'this model': I have lost track of which model you referred to by now.

Gianni et al.: We have corrected this sentence to clarify to which model we are referring (see new lines 172-179).

I. 179; which issue?

Gianni et al.: We have corrected this sentence to clarify that we refer to the origin of the SASWIR subduction-modified mantle anomaly (see new line 180).

I. 186: You're doing everything at the same time here. Build it up layer by layer: you first evaluate whether the anomaly is associated with a slab in the mantle (like Richter, or us in Papua etc). Then you evaluate which subduction zone made that slab, and when. Then you infer how long the anomaly existed, what caused it, etc.

Gianni et al.: The reviewer is right. We have modified this part accordingly (see new lines 180-204).

I. 186-192: Delete the last sentence 'Based on...SASWIR': show the analysis first before you push for the conclusion.

Gianni et al.: We understand the reviewer's point. Nevertheless, we must leave these last sentences because the journal requires the last paragraph of the introduction to be a summary of our findings.

L

. 244-286: 'Also...SASWIR': this section should come later in the paper. First describe the anomaly, then link to tomography, then link the slab to a paleo-subduction zone, then independently confirm with this Karoo link in the Jurassic that a subduction-enriched wedge may already have been present here in the Jurassic.

Gianni et al.: We have discussed with the rest of the coauthors about moving this part, but we decided to keep it where it is. We think that demonstrating since the beginning of the results section that the SASWIR mantle anomaly is a Jurassic feature paves the way towards the tomographic analysis of Jurassic subduction mantle records in the following part of this section.

I. 259: For what did you use UUP07, and for what the vote maps?

Gianni et al.: We now explain this point (see new lines 277).

I. 277: 2150 km and deeper

Gianni et al.: Corrected (see new line 295).

I. 280: at analyzed depths? I don't understand what you mean.

Gianni et al.: This is because assuming average whole-mantle slab sinking rates of 1-1,2 cm/yr at a depth of 2150 km the GI records Jurassic subduction, but at the base the GI records subduction until Permian times (van der Meer et al. 2018) (see new line 300).

I. 287 and onwards: this is a next paragraph, in which you add time to the picture.

Gianni et al.: Corrected (see new line 307).

I. 292: this should be ref 36.

Gianni et al.: Corrected to 38 (former ref. 36) (see new line 312).

I. 302: Does the reconstructed inboard magmatic arc end up above the position of the GI slab?

Gianni et al.: No, the magmatic arc shutted down at ~1200 km from the trench, but the slab tip must have continued migrating inland as occurred in the Laramide flat-slab. This is explained in more detail in the discussion section where we described our model of Fig. 6.

I.305: South Africa: is this the formation of the Cape Fold-belt?

Gianni et al.: Yes, but only the last shortening stage. As reviewed by Navarrete et al. (2019), the Late Triassic flat slab model explains a series of compressional reactivations of permian fold and thrust belts (within them, the Cape fold belt), positive basin inversion of former extensional basins in Patagonia, and ductile shearing in the active Patagonian-Antarctic Peninsula active margin (Riley et al., 2020; Suárez et al., 2020).

I. 311: what kind of alternative reference frames? You need a mantle frame, otherwise you don't have paleolongitudinal control. And for the Jurassic, all you have is van der Meer et al., 2010, Torsvik and Cocks 2017 (you can download the reconstruction from the CEED website), and the recent 'tectonic rules' frame of Muller et al., 2022. No other frames go back far enough as far as I know.

Gianni et al.: The reviewer is totally right. Now, we only use mantle reference frames as suggested and removed our analysis including the model by Müller et al. (2016). Please note that the plate kinematic model of Muller et al. (2019) originally applied in this study (Fig. 3d), uses the mantle reference frame of Tetley et al. (2019). The latter authors presented a method applying a joint global inversion to evaluate the contribution of multiple time-dependent absolute plate motion constraints including fit to age-progressive hotspot tracks, optimizing subduction zone migration behaviors, and minimizing rates of net lithospheric rotation. This method provides both paleo-latitudes and paleo-longitudes relative to the mantle. This approach has been extended in Muller et al. (2022) by including evaluation of continental velocities relative to the mantle as an additional criterion and extending the tectonic rules-based approach proposed by Tetley et al. (2019) to the last billion years. Now, we use the mantle reference frame of Muller et al. (2022) as suggested by the reviewer. Also, we included the plate kinematic model of Torsvik & Cocks (2016) that applies the PGZ method for their mantle reference frame as suggested by the reviewer (See new fig. 3 and new supplementary fig. 3). The spatio-temporal relationship between the mantle and surface geological records of flat subduction is supported by plate kinematic reconstructions applying the different mantle reference frames (Muller et al. 2019 and 2022, van der Meer et al., 2010, Fig. 3d, Supplementary Fig. 3a,b). The only exception is in the reconstruction with the mantle reference frame of Torsvik and Cocks (2016). This presents similar results to the rest of the reconstructions at 220-210 Myr ago but differ since 195 Myr ago, depicting major offsets between the mantle slabs (mainly in the central and northern Pangea margin) and the reconstructed trench (Supplementary Fig. 3c). This observation would indicate intra-oceanic subduction offboard the Southwestern margin of Gondwana at this time, which is not

compatible with the geological records of an Early Jurassic continental arc and back arc basins in western South America. This could be possibly caused by limitations (see Flaments et al., 2022) in the underlying assumptions in the plume generation method used in the reconstructions by Torsvik and Cocks (2016). Thus, we consider Müller et al (2019, 2022) and van der Meer et al. (2010) as the most consistent with the geological records in the active margin of southwestern Gondwana.

L. 324-328: Rather than arguing for this conclusion, explain to the reader what the problem is that we now need to solve if we accept that the SASWIR anomaly is linked to the GI slab, which is linked to Jurassic flat slab subduction.

Gianni et al.: The reviewer is right. We have modified this part as suggested (see new lines 351-358).

I. 331: Flat slabs cannot be caused by slab retreat. They are either advancing, or retreating slower than upper plates, but retreat always makes flat slabs smaller than if there had not been no retreat.

Gianni et al.: 'the reviewer is right. We now indicate that it is a 'forced' trench retreat as suggested by Sheppers et al. (see new lines 370-371).

I. 338: depending on the mantle frame rather than plate model. Also: you use the frames of Muller here, but how did they constrain the paleolongitude before the Cretaceous? I don't think they had paleolongitudinal control. They arguably do in their 'tectonic rules' frame (2022 Solid Earth), so that one would make more sense to use. And the plume-LLSVP-fitted frame of Torsvik & Cocks 2017.

Gianni et al.: The reviewer is right. We have addressed this comment by including the mantle reference frames of Muller et al. (2022) and Torsvik and Cocks (2016) in addition to that of van der Meer et al. (2010) and Müller et al. (2019) (see new Fig. 3a, supplementary fig. 3, and fig. 4). Please note that Muller et al. (2019) shown in fig. 3, already includes the mantle reference frame of Tetley et al. (2019), the first to suggest the tectonic rules-based mantle reference frame. We opted to show reconstructions with the mantle reference frame of Tetley et al. (2019) in fig. 3d because it provides the maximum possible paleo-trench-Georgia islands slab distance from which we derived the kinematics included in our 2-D numerical modeling. Also, note that with the exception of the reconstructions with the mantle reference frame of Torsvik and Cocks (2016), which yield results not compatible with the geological records at 195 Ma (see comment above), the other three reference frames yield very similar paleo-trench-slab distances (22800-2600 km) (Fig. 3d, Supplementary fig. 3, fig. 4).

I. 345: I can see this line of reasoning, but it would help the reader if you first explain what the problem is that you try to solve with this analysis. What are you trying to learn? I mean: once you establish a link between the GI slab and the SASWIR anomaly, the problem to explain the anomaly is solved. You don't need to model flat slabs for that anymore. So the step to the flat slabs is a next layer: you are now using the SASWIR anomaly to learn something about the geochemical traces of flat slab events or so? Please explain what you want to solve with this analysis.

Gianni et al.: The reviewer is right. We have added a sentence at the beginning of this paragraph to highlight to the readers that the location of the Georgia Islands slab and the SASWIR mantle anomaly both far from the reconstructed margins, is telling us something important about how this anomaly was formed so far from the active margin (see new lines 360-365).

I. 375: Depending on the mantle frame, not the plate kinematic model

Gianni et al.: Corrected (see new line 375).

I. 393: A key aspect for what?

Gianni et al.: Corrected (see new line 428).

I. 395: Are you testing the feasibility? Suppose that you are not able to model this flat slab, would you then conclude that the SASWIR anomaly is not related to subduction? I don't think so, right? Aren't you rather using the correlations you made to infer the conditions under which the flat subduction occurred? Wouldn't that be more interesting? I'm not sure what it is you want to learn through the modeling.

Gianni et al.: The reviewer is right. We have modified these sentences accordingly (see new lines 432-435).

L. 401: I think it's the other way around. You reconstructed THAT the geochemical anomaly was transported far inboard, and still remains vertically above the associated slab. So now you have to 'train' your model to have that as a result, and you can learn about the dynamics of very large flat slabs. Or so.

Gianni et al.: We agree with the reviewer. This sentence has been modified accordingly (see new lines 432-440).

I. 417: As presented, you just add a numerical 'cartoon' of a model you already interpreted before the modeling started. I think you can do more with the modeling, to learn about conditions you cannot reconstruct (rheology, whatever). You can leave it as is, but it doesn't add much.

Gianni et al.: We are using the 2-D numerical model to understand if the flat-slab phenomenon of more than 2000 km is thermodynamically viable under the obvious limitations of a numerical model. We are interested in studying the conditions under which a flat-slab of these characteristics is viable. We have done extensive testing to evaluate rheological and thermal parameters to constrain those variables within previously reported value frames. It is clear that it is interesting to carry out a meticulous analysis of the behavior of the flat-slab by modifying these variables (which we did and we are working to publish the results) and gain deeper insights into the causes of large-scale flat subduction, but it seemed to us that it did not fall within the scope of this work.

I. 437: Would that give the same signature as an arc/mantle wedge?

Gianni et al.: Yes, if it was metasomatized by slab fluids. We added this in the sentence to clarify this point (see new lines 481-483).

I. 446: detritus is something I'd associate with a sandstone. Wouldn't 'fragments' be a better word?

Gianni et al.: The reviewer is right. This is now corrected (see new lines 492).

I. 457: well, basically, the explanation for the anomaly is the same in this scenario and in the previous ones, but the prediction for mantle flow is very different. In the former explanations, the uppermost mantle would flow opposite to plate motion. What would drive that?

Gianni et al.: Now we indicate that this model implies an eastward upper mantle flow of subduction-influenced asthenosphere that opposes continental plate motion in the study area, which is not easy to explain from a geodynamic point of view (see new lines 475-480).

I. 464: Explain what the problem is: the Gondwana margin was located far to the west of the SASWIR.

Gianni et al.: Corrected (see new lines 507-512).

I. 494: 'inland' is a bit strange here. 'Below the supercontinent'? Also: and transported to more than 2200 km etc

Gianni et al.: Corrected (see new lines 542-545).

I. 561: this term 'asthenospheric anomaly telescoping' is not very helpful to explain what's going on. I don't understand what it means, or why it's important. You just bulldozed mantle wedge eastward with a flat slab.

Gianni et al.: This concept means that a relict subduction-related mantle must not necessarily lie beneath a reconstructed active margin as suggested previously. The telescoping of asthenospheric anomalies suggest that relict subduction-influenced mantle, whether beneath the active margin or beneath intraplate areas (e.g., suture zones), can be jointly accumulated beneath intraplate interiors far from the plate margin creating long-term mantle heterogeneities.

I. 575: why would flat slab subduction result in slab break-off? I'd argue the opposite. You first need to roll that entire slab back, and then it'll break off. Or not (e.g., Boschman et al., G-Cubed 2018).

Gianni et al.: Not always. Some flat-slabs present arc migration patterns that depart from typical trenchward magmatic migration expected during slab steepening and roll back. This is a nice study discussing this issue: Dai, L., Wang, L., Lou, D., Li, Z. H., Dong, H., Ma, F., ... & Yu, S. (2020). Slab rollback versus delamination: Contrasting fates of flat-slab subduction and implications for South China evolution in the Mesozoic. *Journal of Geophysical Research: Solid Earth*, 125(4), e2019JB019164.

I. 543-581: This section contains a series of more or less loose thoughts, but how is this linked to the SASWIR anomalies? What do we really learn from your correlations?

Gianni et al.: We understand this point of view. Nevertheless, we have decided to keep this section as it is. From our point of view, this section summarizes three important aspects that we have learned from this study: 1) the South Gondwana flat slab is likely the largest documented flat subduction event, 2) we note that inland bulldozing of asthenospheric material to create long-lasting near-stationary subduction-related mantle anomaly is an unconsidered geodynamic consequence of flat slabs, and hence, we suggest a name for this geodynamic process, and 3) as flat subduction is common in active subduction settings, and likely was more common in the past, this must have been an important contributor to the development of upper mantle heterogeneities beneath continental interiors.

l. 591: well, this is not a speculation, but a deduction. If you have a neutrally buoyant wedge in the mantle, and it's not moving, then the mantle is not moving. You can't have one without the other.

Gianni et al.: Corrected (see new line 648).

l. 604: this is the same point as you made before.

Gianni et al.: We have modified this sentence (see new lines 655-661).

We are deeply thankful to Dr. J. J. Van Hinsbergen for his thorough, constructive, and kind revision of our manuscript.

Guido M. Gianni, Jeremías Likerman, César R. Navarrete, Conrado R. Gianni, and Sergio Zlotnik